# Dual Parameterization of Sparse Variational Gaussian Processes

**Vincent Adam**[*]
Aalto University / Secondmind.ai
Espoo, Finland / Cambridge, UK
vincent.adam@aalto.fi

**Paul E. Chang**[*]
Aalto University
Espoo, Finland
paul.chang@aalto.fi

**Mohammad Emtiyaz Khan**
RIKEN Center for AI Project
Tokyo, Japan
emtiyaz.khan@riken.jp

**Arno Solin**
Aalto University
Espoo, Finland
arno.solin@aalto.fi

## Abstract

Sparse variational Gaussian process (SVGP) methods are a common choice for non-conjugate Gaussian process inference because of their computational benefits. In this paper, we improve their computational efficiency by using a dual parameterization where each data example is assigned dual parameters, similarly to site parameters used in expectation propagation. Our dual parameterization speeds-up inference using natural gradient descent, and provides a tighter evidence lower bound for hyperparameter learning. The approach has the same memory cost as the current SVGP methods, but it is faster and more accurate.

## 1   Introduction

Gaussian processes (GPs, [31]) have become ubiquitous models in the probabilistic machine learning toolbox, but their application is challenging due to two issues: poor $\mathcal{O}(n^3)$ scaling in the number of data points, $n$, and challenging approximate inference in non-conjugate (non-Gaussian) models. In recent years, variational inference has become the go-to solution to overcome these problems, where sparse variational GP methods [35] tackle both the non-conjugacy and high computation cost. The computation is reduced to $\mathcal{O}(nm^2)$ by using a small number of $m \ll n$ inducing-input locations. For large problems, the SVGP framework is a popular choice as it enables fast stochastic training, and reduces the cost to $\mathcal{O}(m^3 + n_\mathrm{b}m^2)$ per step, where $n_\mathrm{b}$ is the batch size [10, 11].

It is a common practice in SVGP to utilize the standard mean-covariance parameterization which requires $\mathcal{O}(m^2)$ memory. Inference is carried out by optimizing an objective $\mathcal{L}(q)$ that uses a Gaussian distribution $q$ parameterized by parameters $\boldsymbol{\xi} = (\mathbf{m}, \mathbf{L})$ where $\mathbf{m}$ is the mean and $\mathbf{L}$ is the Cholesky factor of the covariance matrix. We will refer to the SVGP methods using such parameterization as the $q$-SVGP methods. The advantage of this formulation is that, for log-concave likelihoods, the objective is convex and gradient-based optimization works well [2]. The optimization can further be improved by using natural gradient descent (NGD) which is shown to be less sensitive to learning rates [33]. The NGD algorithm with $q$-SVGP parameterization is currently the state-of-the-art and available in the existing software implementations such as GPflow [26] and GPyTorch [9].

An alternate parameterization to the $q$-SVGP parameterization is the one where every likelihood is assigned two sets of parameters which require $\mathcal{O}(n)$ memory. Existence of such parameterizations was initially shown by Csató and Opper [8] for general GP models, and later on extended to variational

---

[*]Both authors contributed equally.

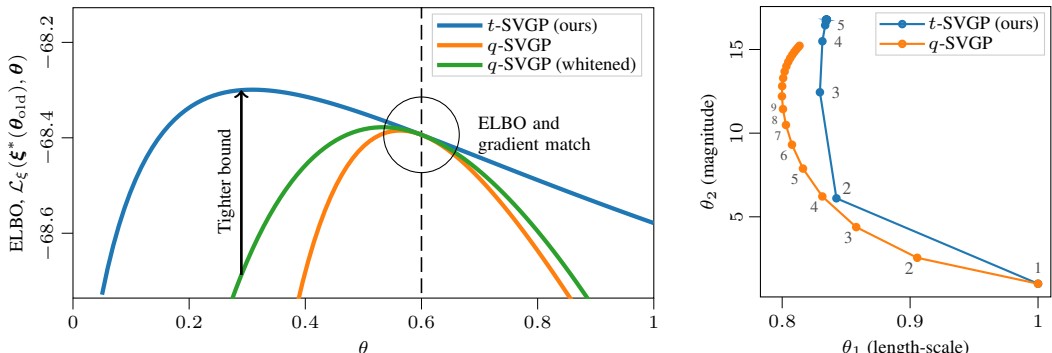

Figure 1: The dual parameterization gives a tighter bound compared to standard SVGP (whitened/unwhitened) as shown on the left for a GP classification tasks and varying kernel magnitude $\theta$. The tighter bound helps take longer steps, speeding up convergence of hyperparameters, as shown on the right for the *banana classification* task with coordinate ascent w.r.t. $\boldsymbol{\theta}$ and variational parameters $\boldsymbol{\xi}$.

objectives for GPs [28, 29], and also to latent Gaussian models by using Lagrangian duality [20, 16]. Due to this later connection, we refer to this parameterization as the *dual parameterization* where the parameters are the Lagrange multipliers, ensuring that the marginal mean and variance of each latent function is consistent to the marginals obtained by the full GP [16]. Expectation propagation (EP, [27]) too naturally employ such parameterizations, but by using site parameters. Although unrelated to duality, such methods are popular for GP inference, and due to this connection, we will refer to the methods using dual parameterization as $t$-SVGP (the letter '$t$' refers to the sites). To the best of our knowledge, the dual parameterization for SVGP has only been used to speed up computation and inference for the specific case of Markovian GPs [4, 38].

The main contribution of this work is to introduce the dual parameterization for SVGP and show that it speeds up both the learning and inference. For inference, we show that the dual parameters are automatically obtained through a different formulation of NGD, written in terms of the expectation parameters [17, 15, 18]. The formulation is fast since it avoids the use of sluggish automatic differentiation to compute the natural gradients. We also match the typical $\mathcal{O}(m^2)$ memory complexity in other SVGP methods by introducing a *tied* parametrization. For learning, we show that the dual parameterization results in a *tighter* lower bound to the marginal likelihood, which speed-up the hyperparameter optimization (see Fig. 1). We provide extensive evaluation on benchmark data sets, which confirms our findings. Our work attempts to revive the dual parameterization, which was popular in the early 2000s, but was somehow forgotten and not used in the recent SVGP algorithms.

## 2  Background: Variational Inference for Gaussian Processes Models

Gaussian processes (GPs, [31]) are distributions over functions, commonly used in machine learning to endow latent functions in generative models with rich and interpretable priors. These priors can provide strong inductive biases for regression tasks in the small data regime. GP-based models are the ones that employ a GP prior over the (latent) functions $f(\cdot) \sim \mathcal{GP}(\mu(\cdot), \kappa(\cdot, \cdot'))$, where the prior is completely characterized by the mean function $\mu(\cdot)$ and covariance function $\kappa(\cdot, \cdot')$. We denote $f(\cdot)$ as a function but occasionally simplify the notation to just $f$ in the interest of reducing clutter. Given a data set $\mathcal{D} = (\mathbf{X}, \mathbf{y}) = \{(\mathbf{x}_i, y_i)\}_{i=1}^n$ of input–output pairs, we denote by $\mathbf{f}$ the vector of function evaluations at the inputs $\{f(\mathbf{x}_i)\}_{i=1}^n$. The function evaluation at $\mathbf{x}_i$ are passed through likelihood functions to model the outputs $y \in \mathbb{R}$, *e.g.* by specifying $p(\mathbf{y} \mid \mathbf{f}) \coloneqq \prod_{i=1}^n p(y_i \mid f_i)$. Prediction at a new test input $\mathbf{x}_*$ is obtained by computing the distribution $p(f(\mathbf{x}_*) \mid \mathcal{D}, \mathbf{x}_*)$. For Gaussian likelihoods $N(y_i \mid f_i, \sigma^2)$, the predictive distribution is available in closed form as a Gaussian distribution $N(f(\mathbf{x}_*) \mid m_{\text{GPR}}(\mathbf{x}_*), v_{\text{GPR}}(\mathbf{x}_*))$ with mean and variance defined as follows:

$$m_{\text{GPR}}(\mathbf{x}_*) \coloneqq \mathbf{k}_{\mathbf{f}*}^\top (\mathbf{K}_{\mathbf{ff}} + \sigma^2 \mathbf{I}_n)^{-1} \mathbf{y}, \text{ and } v_{\text{GPR}}(\mathbf{x}_*) \coloneqq \kappa_{**} - \mathbf{k}_{\mathbf{f}*}^\top (\mathbf{K}_{\mathbf{ff}} + \sigma^2 \mathbf{I}_n)^{-1} \mathbf{k}_{\mathbf{f}*}, \quad (1)$$

where $\mathbf{k}_{\mathbf{f}*}$ is a vector of $\kappa(\mathbf{x}_*, \mathbf{x}_i)$ as the $i^{\text{th}}$ element for all $\mathbf{x}_i \in \mathbf{X}$, $\mathbf{K}_{\mathbf{ff}}$ is an $n \times n$ matrix with $\kappa(\mathbf{x}_i, \mathbf{x}_j)$ as the $ij^{\text{th}}$ entry, and $\kappa_{**} = \kappa(\mathbf{x}_*, \mathbf{x}_*)$.

## 2.1 Variational Expectation–Maximization for GPs with Non-Conjugate Likelihoods

For non-Gaussian likelihoods, the posterior and predictive distributions are no longer Gaussian, and we need to resort to approximate inference methods. Variational inference is a popular choice because it allows for fast posterior approximation and hyperparameter learning via stochastic training [35, 10]. Denoting kernel hyperparameters by $\boldsymbol{\theta}$ and the corresponding GP prior by $p_{\boldsymbol{\theta}}(\mathbf{f})$, the posterior distribution can be written as $p_{\boldsymbol{\theta}}(\mathbf{f} \mid \mathbf{y}) = p_{\boldsymbol{\theta}}(\mathbf{f}) \, p(\mathbf{y} \mid \mathbf{f})/p_{\boldsymbol{\theta}}(\mathbf{y})$, where $p_{\boldsymbol{\theta}}(\mathbf{y}) = \int p_{\boldsymbol{\theta}}(\mathbf{f}, \mathbf{y})\mathrm{d}\mathbf{f}$ is the marginal likelihood of the observations. We seek to approximate $p_{\boldsymbol{\theta}}(\mathbf{f} \mid \mathbf{y}) \approx q_{\mathbf{f}}(\mathbf{f})$ by a Gaussian distribution whose parameters can be obtained by optimizing the following evidence lower bound (ELBO) to the log-marginal likelihood,

$$\log p_{\boldsymbol{\theta}}(\mathbf{y}) \geq \mathcal{L}_q(q_{\mathbf{f}}, \boldsymbol{\theta}) = \sum_{i=1}^{n} \mathbb{E}_{q_{\mathbf{f}}(f_i)} \left[ \log p(y_i \mid f_i) \right] - \mathrm{D}_{\mathrm{KL}}[q_{\mathbf{f}}(\mathbf{f}) \, \| \, p_{\boldsymbol{\theta}}(\mathbf{f})] . \quad (2)$$

For the variational approximation, it is a standard practice to choose the mean-covariance parameterization, denoted by $\boldsymbol{\xi} = (\mathbf{m}, \mathbf{S})$. It is also common to use the Cholesky factor $\mathbf{L}$ instead [3] since it is uniquely determined for a covariance matrix. The multivariate normal distribution is part of the exponential family [37], i.e. it's probability density function take the form $p(\mathbf{x}) = \exp(\boldsymbol{\eta}^{\top} \mathbf{T}(\mathbf{x}) - a(\boldsymbol{\eta}))$, with natural parameters $\boldsymbol{\eta} = (\mathbf{S}^{-1}\mathbf{m}, -\mathbf{S}^{-1}/2)$ and sufficient statistics $\mathbf{T}(\mathbf{x}) = [\mathbf{x}, \mathbf{x}\mathbf{x}^{\top}]$. This natural parameterization is also a common choice, along with the associated expectation parameterization $\boldsymbol{\mu} = \mathbb{E}_q[\mathbf{T}(\mathbf{x})] = (\mathbf{m}, \mathbf{S} + \mathbf{m}\mathbf{m}^{\top})$. A final choice of the parameterization, called the *whitened* parameterization [36], uses a variable $\mathbf{v} \sim \mathrm{N}(\mathbf{v}; \mathbf{m}_{\mathbf{v}}, \mathbf{S}_{\mathbf{v}})$ along with the transformation $\mathbf{f} = \mathbf{L}\mathbf{v}$, to parameterize $\mathbf{m} = \mathbf{L}\mathbf{m}_{\mathbf{v}}$ and $\mathbf{S} = \mathbf{L}\mathbf{S}_{\mathbf{v}}\mathbf{L}^{\top}$.

In the following, we will consider the ELBO with several parameterizations and, to make the notation clearer, we will indicate the parameterization used with a subscript with $\mathcal{L}$. For example, we may have $\mathcal{L}_{\xi}(\boldsymbol{\xi}, \boldsymbol{\theta}) = \mathcal{L}_{\eta}(\boldsymbol{\eta}, \boldsymbol{\theta}) = \mathcal{L}_{\mu}(\boldsymbol{\mu}, \boldsymbol{\theta})$, which are all clearly equal due to a unique mapping between the parameterizations; see [25] for more details on the maps.

The ELBO can be optimized by using a variational expectation–maximization (VEM) procedure, where we alternate between optimizing variational parameters, say $\boldsymbol{\xi}$, and hyperparameters $\boldsymbol{\theta}$,

$$\text{E-step:} \quad \boldsymbol{\xi}_t^* \leftarrow \arg\max_{\boldsymbol{\xi}} \mathcal{L}_{\xi}(\boldsymbol{\xi}, \boldsymbol{\theta}_t), \qquad \text{M-step:} \quad \boldsymbol{\theta}_{t+1} \leftarrow \arg\max_{\boldsymbol{\theta}} \mathcal{L}_{\xi}(\boldsymbol{\xi}_t^*, \boldsymbol{\theta}), \quad (3)$$

where $t$ denotes the iterations, and we have explicitly written the dependence of optimal parameter $\boldsymbol{\xi}^*(\boldsymbol{\theta}_t)$ as a function of the old parameter $\boldsymbol{\theta}_t$. Both and E and M-steps can be carried out with gradient descent, for example, using an iteration of the form $\boldsymbol{\xi}_t^{(k+1)} \leftarrow \boldsymbol{\xi}_t^{(k)} + \rho_k \nabla_{\boldsymbol{\xi}} \mathcal{L}_{\xi}(\boldsymbol{\xi}_t^{(k)}, \boldsymbol{\theta}_t)$ for E-step, which would ultimately converge to $\boldsymbol{\xi}_t^*$. A similar iterative method can be used for the M-step.

## 2.2 Inference via Natural-Gradient Descent (NGD)

A popular strategy for the E-step is to use natural gradient descent where we replace the gradient by the one preconditioned using the Fisher information matrix $\mathbf{F}(\boldsymbol{\xi})$ of $q_{\mathbf{f}}(\mathbf{f})$. We denote natural gradients by $\tilde{\nabla}_{\boldsymbol{\xi}} \mathcal{L}_{\xi}(\boldsymbol{\xi}, \boldsymbol{\theta}) = \mathbf{F}(\boldsymbol{\xi})^{-1} \nabla_{\boldsymbol{\xi}} \mathcal{L}_{\xi}(\boldsymbol{\xi}, \boldsymbol{\theta})$, to get the following update,

$$\boldsymbol{\xi}_t^{(k+1)} \leftarrow \boldsymbol{\xi}_t^{(k)} + \rho_k \tilde{\nabla}_{\boldsymbol{\xi}} \mathcal{L}_{\xi}(\boldsymbol{\xi}_t^{(k)}, \boldsymbol{\theta}_t). \quad (4)$$

Such updates can converge faster than gradient descent [33, 17, 18], and at times are less sensitive to the choice of the learning rate $\rho_k$ due to the scaling with the Fisher information matrix. The implementation simplifies greatly when using natural parameterizations,

$$\boldsymbol{\eta}_t^{(k+1)} \leftarrow \boldsymbol{\eta}_t^{(k)} + \rho_k \nabla_{\boldsymbol{\mu}} \mathcal{L}_{\mu}(\boldsymbol{\mu}_t^{(k)}, \boldsymbol{\theta}_t), \quad (5)$$

because $\tilde{\nabla}_{\boldsymbol{\eta}} \mathcal{L}_{\eta}(\boldsymbol{\eta}, \boldsymbol{\theta}) = \nabla_{\boldsymbol{\mu}} \mathcal{L}_{\mu}(\boldsymbol{\mu}, \boldsymbol{\theta})$, that is, the natural gradients with respect to $\boldsymbol{\eta}$ are in fact the gradients with respect to the expectation parameter $\boldsymbol{\mu}$ [18]. In the remainder of the paper we will frequently use this property, and refer to the natural gradient with respect to $\boldsymbol{\eta}$ by the gradients with respect to expectation parameterization $\boldsymbol{\mu}$. The VEM procedure with NGD has recently become a popular choice for sparse variants of GPs, which we explain next.

## 2.3 Sparse Variational GP Methods and Their Challenges

Inference in GP models, whether conjugate or non-conjugate, suffers from an $\mathcal{O}(n^3)$ computational bottleneck required to invert the posterior covariance matrix. A common approach to reduce the

computational complexity is to use a sparse approximation relying on a small number $m \ll n$ representative inputs, also called *inducing* inputs, denoted by $\mathbf{Z} \coloneqq (\mathbf{z}_1, \mathbf{z}_2, \ldots, \mathbf{z}_m)$ [34, 7, 30, 39]. Sparse variational GP methods [35, 10, 11, 5, 6, 32] rely on a Gaussian approximation $q_{\mathbf{u}}(\mathbf{u})$ over the functions $\mathbf{u} = (f(\mathbf{z}_1), f(\mathbf{z}_2), \ldots, f(\mathbf{z}_m))$ to approximate the posterior over arbitrary locations,

$$q_{\mathbf{u}, \boldsymbol{\theta}}(f(\cdot)) = \int p_{\boldsymbol{\theta}}(f(\cdot) \mid \mathbf{u}) \, q_{\mathbf{u}}(\mathbf{u}) \, \mathrm{d}\mathbf{u}, \qquad (6)$$

where $p_{\boldsymbol{\theta}}(f(\cdot) \mid \mathbf{u})$ is a conditional of the GP prior. For example, for a Gaussian $q_{\mathbf{u}}(\mathbf{u}) = \mathrm{N}(\mathbf{u}; \mathbf{m}_{\mathbf{u}}, \mathbf{S}_{\mathbf{u}})$, the posterior marginal of $f_i = f(\mathbf{x}_i)$ takes the following form,

$$q_{\mathbf{u}, \boldsymbol{\theta}}(f_i) = \mathrm{N}\left(f_i \mid \mathbf{a}_i^\top \mathbf{m}_{\mathbf{u}}, \kappa_{ii} - \mathbf{a}_i^\top (\mathbf{K}_{\mathbf{uu}} - \mathbf{S}_{\mathbf{u}}) \mathbf{a}_i\right), \qquad (7)$$

where $\mathbf{a}_i = \mathbf{K}_{\mathbf{uu}}^{-1} \mathbf{k}_{\mathbf{u}i}$ with $\mathbf{K}_{\mathbf{uu}}$ as the prior covariance evaluated at $\mathbf{Z}$, and $\mathbf{k}_{\mathbf{u}i}$ as an $m$-length vector of $\kappa(\mathbf{z}_j, \mathbf{x}_i), \forall j$. The parameters $\boldsymbol{\xi}_{\mathbf{u}} = (\mathbf{m}_{\mathbf{u}}, \mathbf{S}_{\mathbf{u}})$ can be learned via an ELBO similar to Eq. (2),

$$\mathcal{L}_\xi(\boldsymbol{\xi}_{\mathbf{u}}, \boldsymbol{\theta}) \coloneqq \sum_{i=1}^n \mathbb{E}_{q_{\mathbf{u}, \boldsymbol{\theta}}(f_i)} \left[\log p(y_i \mid f_i)\right] - \mathrm{D}_{\mathrm{KL}}[q_{\mathbf{u}}(\mathbf{u}) \,\|\, p_{\boldsymbol{\theta}}(\mathbf{u})]. \qquad (8)$$

The variational objective for such sparse GP posteriors can be evaluated at a cost $\mathcal{O}(nm^2 + m^3)$, and optimization can be performed in $\mathcal{O}(m^3 + n_{\mathrm{b}} m^2)$ per iteration via stochastic natural-gradient methods with mini-batch size $n_{\mathrm{b}}$ [10]. This formulation also works for general likelihood functions [11]. This and the low computational complexity has lead to a wide adoption of the SVGP algorithm. It is currently the state-of-the-art for sparse variants of GP and is available in the existing software implementations such as GPflow [26] and GPyTorch [9].

Despite their popularity, the current implementations are cumbersome and there is plenty of room for improvements. For example, the methods discussed in Salimbeni et al. [33] (see App. D for a summary) rely on the mean-covariance parameterization and NGD is performed in $\boldsymbol{\eta}$-space. However, the natural gradients are implemented via chain rule: $(\nabla_{\boldsymbol{\mu}} \boldsymbol{\xi}) \nabla_{\boldsymbol{\xi}} \mathcal{L}_\xi(\boldsymbol{\xi}, \boldsymbol{\theta})$ utilizing the gradients in the $\boldsymbol{\xi}$-space, which requires computation of additional Jacobians and multiplication operations. Since other operations are done via mean-covariance parameterization, we need to go back and forth between $\boldsymbol{\eta}$ and $\boldsymbol{\xi}$, which further increases the cost. In addition, many existing implementations currently compute the natural gradient of the *whole* ELBO, including the KL term which is not required; see Khan and Rue [19, Sec. 2.2]. Finally, the M-step is dependent on the choice of the parameterization used in E-step and can affect the convergence speed. To the best of our knowledge, this has not been investigated in the literature.

In what follows, we argue to use a *dual* parameterization instead of the usual mean-covariance parameterization, and show that this not only simplifies computations of natural-gradients, but also gives rise to a tighter bound for hyperparameter learning and speed-up the whole VEM procedure.

## 3 The $t$-VGP Method: Dual-Parameter Based Learning for GPs

We start with a property of the optimal $q_{\mathbf{f}}^*$ of Eq. (2). Khan and Nielsen [18, Eq. (18)] show, that $q_{\mathbf{f}}^*$ can be parameterized by 2D vectors $\boldsymbol{\lambda}_i^* = (\lambda_{1,i}^*, \lambda_{2,i}^*)$ used in site functions $t_i^*(f_i)$,

$$q_{\mathbf{f}}^*(\mathbf{f}) \propto p_{\boldsymbol{\theta}}(\mathbf{f}) \prod_{i=1}^n \underbrace{e^{\langle \boldsymbol{\lambda}_i^*, \mathbf{T}(f_i) \rangle}}_{t_i^*(f_i)}, \quad \text{where } \boldsymbol{\lambda}_i^* = \nabla_{\boldsymbol{\mu}_i} \mathbb{E}_{q_{\mathbf{f}}^*(f_i)}[\log p(y_i \mid f_i)]. \qquad (9)$$

The vectors $\boldsymbol{\lambda}_i^*$ are equal to the natural gradient of the expected log-likelihood, where the expectation is taken with respect to the posterior marginal $q_{\mathbf{f}}^*(f_i) = \mathrm{N}(f_i; m_i^*, S_{ii}^*)$ with $S_{ii}^*$ as the $i$'th diagonal element of $\mathbf{S}^*$, and the gradient is taken with respect to its expectation parameter $\boldsymbol{\mu}_i = (m_i, m_i^2 + S_{ii})$ and evaluated at $\boldsymbol{\mu}_i^*$. The vector $\mathbf{T}(f_i) = (f_i, f_i^2)$ are the sufficient statistics of a Gaussian. The $q_{\mathbf{f}}^*$ uses local unnormalized Gaussian *sites* $t_i^*(f_i)$, similarly to those used in the Expectation Propagation (EP) algorithm [27]. The difference here is that the site parameters $\boldsymbol{\lambda}_i^*$ are equal to natural gradients of the expected log-likelihood which are easy to compute using the gradient with respect to $\boldsymbol{\mu}_i$.

The parameters $\boldsymbol{\lambda}_i^*$ can be seen as the optimal *dual* parameters of a Lagrangian function with moment-matching constraints. We can show this in two steps:

1. For each $p(y_i | f_i)$, we introduce a *local* Gaussian $\widetilde{q}_i(f_i; \widetilde{\boldsymbol{\mu}}_i)$ with expectation parameters $\widetilde{\boldsymbol{\mu}}_i$.

2. Then, we aim to match $\widetilde{\boldsymbol{\mu}}_i$ with the marginal moments $\boldsymbol{\mu}_i$ of the *global* Gaussian $q_{\mathbf{f}}(\mathbf{f}; \boldsymbol{\mu})$.

This is written below as a Lagrangian where the middle term (shown in red) 'decouples' the terms using the local Gaussians from those using the global Gaussian,

$$\mathcal{L}_{\text{Lagrange}}(\boldsymbol{\mu}, \widetilde{\boldsymbol{\mu}}, \boldsymbol{\lambda}) = \sum_{i=1}^{n} \mathbb{E}_{\widetilde{q}_i(f_i; \widetilde{\boldsymbol{\mu}}_i)}[\log p(y_i \mid f_i)] - \sum_{i=1}^{n} \langle \boldsymbol{\lambda}_i, \widetilde{\boldsymbol{\mu}}_i - \boldsymbol{\mu}_i \rangle - \mathrm{D}_{\text{KL}}[q_{\mathbf{f}}(\mathbf{f}; \boldsymbol{\mu}) \,\|\, p_{\boldsymbol{\theta}}(\mathbf{f})]. \tag{10}$$

The parameter $\boldsymbol{\lambda}_i$ is the Lagrange multiplier of the moment-matching constraint $\widetilde{\boldsymbol{\mu}}_i = \boldsymbol{\mu}_i$ (here, $\boldsymbol{\lambda}$ and $\widetilde{\boldsymbol{\mu}}$ denote the sets containing all $\boldsymbol{\lambda}_i$ and $\widetilde{\boldsymbol{\mu}}_i$). The optimal $\boldsymbol{\lambda}_i$ is equal to $\boldsymbol{\lambda}_i^*$ shown in Eq. (9). We can show this by, first setting the derivative with respect to $\boldsymbol{\lambda}_i$ to 0, finding that the constraints $\widetilde{\boldsymbol{\mu}}_i^* = \boldsymbol{\mu}_i^*$ are satisfied. Using this and by setting the derivative with respect to $\widetilde{\boldsymbol{\mu}}_i$ to 0, we see that the optimal $\boldsymbol{\lambda}_i^*$ is in fact equal to the natural gradients, as depicted in Eq. (9).

The optimal natural parameter of $q_{\mathbf{f}}^*(\mathbf{f})$, denoted by $\boldsymbol{\eta}^*$, has an 'additive' structure that, as we will show, can be exploited to speed-up learning. The structure follows by setting derivatives w.r.t. $\boldsymbol{\mu}$ to 0,

$$\nabla_{\boldsymbol{\mu}} \mathrm{D}_{\text{KL}}[q_{\mathbf{f}}(\mathbf{f}; \boldsymbol{\mu}^*) \,\|\, p_{\boldsymbol{\theta}}(\mathbf{f})] = \boldsymbol{\lambda}^* \quad \implies \quad \boldsymbol{\eta}^* = \boldsymbol{\eta}_0(\boldsymbol{\theta}) + \boldsymbol{\lambda}^*. \tag{11}$$

The second equality is obtained by using the result that $\nabla_{\boldsymbol{\mu}} \mathrm{D}_{\text{KL}}[q(\mathbf{f}; \boldsymbol{\mu}) \,\|\, p_{\boldsymbol{\theta}}(\mathbf{f})] = \boldsymbol{\eta} - \boldsymbol{\eta}_0(\boldsymbol{\theta})$ where $\boldsymbol{\eta}$ and $\boldsymbol{\eta}_0(\boldsymbol{\theta})$ are natural parameters of $q(\mathbf{f}; \boldsymbol{\mu})$ and $p_{\boldsymbol{\theta}}(\mathbf{f})$ respectively [19, Sec. 2.2]. The final result follows by noting that the right-hand side is the natural parameter of $q_{\mathbf{f}}^*(\mathbf{f})$ from Eq. (9). This implies that the global $q_{\mathbf{f}}(\mathbf{f}; \boldsymbol{\mu}^*) = q_{\mathbf{f}}^*(\mathbf{f})$ is also equal to the optimal approximation, as desired.

The Lagrangian formulation is closely related to the maximum-entropy principle [13] which forms the foundations of Bayesian inference [14]. Through moment matching, the prior is modified to obtain posterior approximations that explain the data well. Since $\boldsymbol{\lambda}_i^*$ are the optimal Lagrange multipliers, they measure the sensitivity of the optimal $q_{\mathbf{f}}^*(\mathbf{f})$ to the perturbation in the constraints, and reveal the relative importance of data examples. Therefore, the structure of the solution $\boldsymbol{\eta}^*$ shown in Eq. (11) is useful for estimating $\boldsymbol{\theta}$. The additive structure can be used to measure the relative importance of the prior to the dual parameters $\boldsymbol{\lambda}_i^*$. Our main idea is to use the structure to speed-up learning for SVGPs.

Eq. (11) can be rewritten in terms of the mean-covariance parameterization, to gain further insight about the structure. To do so, we use Bonnet and Price's theorem, and rearrange to get the following (see Eqs. 10 and 11 in [19] for a similar derivation),

$$\mathbf{m}^* = -\mathbf{K}_{\mathbf{ff}} \boldsymbol{\alpha}^*, \qquad \text{where } \boldsymbol{\alpha}^* \text{ is a vector of } \alpha_i^* = \mathbb{E}_{q_{\mathbf{f}}^*(f_i)} [\nabla_f \log p(y_i \mid f_i)], \tag{12}$$

$$(\mathbf{S}^*)^{-1} = \mathbf{K}_{\mathbf{ff}}^{-1} + \mathrm{diag}(\boldsymbol{\beta}^*), \quad \text{where } \boldsymbol{\beta}^* \text{ is a vector of } \beta_i^* = \mathbb{E}_{q_{\mathbf{f}}^*(f_i)} [-\nabla_{ff}^2 \log p(y_i \mid f_i)]. \tag{13}$$

The variables $\alpha_i^*$ and $\beta_i^*$ can be easily obtained by using the gradient and Hessian of the log-likelihood, and using those we can get $\boldsymbol{\lambda}_i^* = (\beta_i^* m_i^* + \alpha_i^*, -\frac{1}{2} \beta_i^*)$.

Several other works have discussed such parameterizations, although our work is the first to connect it to natural gradients as the optimal Lagrange multiplier. The representation theorem by Kimeldorf and Wahba [22] is perhaps the most general result, but Csató and Opper [8] were the first to derive such parameterization for GPs; see Lemma 1 in their paper. Their result is for *exact* posteriors which is intractable while ours is for Gaussian approximations and easy to compute. A minor difference there is that their parameterizations use the integrals of likelihoods (instead of log-likelihoods) with respect to the GP prior (instead of the posterior), but we can also express them as Eq. (9) where $q_{\mathbf{f}}^*(\mathbf{f})$ is replaced by the true posterior $p_{\boldsymbol{\theta}}(\mathbf{f} \mid \mathbf{y})$.

Parameterization of the variational posterior similar to ours are discussed in [28, 29], but the one by Khan et al. [20] is the most similar. They establish the first connection to duality for cases where ELBO is convex with respect to the mean-covariance parameterization. Khan [16] extends this to non-convex ELBO using the Lagrangian function similar to ours, but written with the mean-covariance parameterization to get the solutions shown in Eqs. (12) and (13). As shown earlier, their $(\boldsymbol{\alpha}, \boldsymbol{\beta})$ parameterization is just a reparameterization of our $\boldsymbol{\lambda}$ parameterization. Here, we argue in favour of our formulation which enables the reformulation in terms of site functions in Eq. (9) and also allows us to exploit the 'additive' structure in Eq. (11) to speed up hyperparameter learning. The mean-covariance parameterization does not have these features.

### 3.1 Improved Objective for Hyperparameter Learning

We will now discuss a method to speed-up VEM by using the dual parameterization. The key idea is to exploit the form given in Eq. (11) to propose a better objective for the M-step.

Standard VEM procedures, such as those shown in Eq. (3), iterate pairs of E and M steps which we here describe in the context of the dual parameterization. In the E-step, starting from a hyperparameter

$\boldsymbol{\theta}_t$, the optimal variational distribution $q_{\mathbf{f}}^*(\mathbf{f})$ maximizing the ELBO in Eq. (2) is computed. For the dual parameterization, we get the optimal variational parameters $\boldsymbol{\eta}_t^* = \boldsymbol{\eta}_0(\boldsymbol{\theta}_t) + \boldsymbol{\lambda}_t^*$. Here, the subscripts $t$ in $\boldsymbol{\eta}_t^*$ and $\boldsymbol{\lambda}_t^*$ indicate the dependence of the E-step iterations on $\boldsymbol{\theta}_t$, while $\boldsymbol{\eta}_0(\boldsymbol{\theta}_t)$ indicates a direct dependence of the prior natural parameter over $\boldsymbol{\theta}_t$. The standard M-step would then be to use $\boldsymbol{\eta}_t^*$ in the ELBO in Eq. (3) as shown below, while we propose an alternate procedure where the prior $\boldsymbol{\eta}_0(\boldsymbol{\theta})$ is left free (shown in red):

$$\text{Standard M-step:} \quad \boldsymbol{\theta}_{t+1} = \arg\min_{\boldsymbol{\theta}} \mathcal{L}_\eta(\boldsymbol{\eta}_0(\boldsymbol{\theta}_t) + \boldsymbol{\lambda}_t^*, \boldsymbol{\theta}) \tag{14}$$

$$\text{Proposed M-step:} \quad \boldsymbol{\theta}_{t+1} = \arg\min_{\boldsymbol{\theta}} \mathcal{L}_\eta(\boldsymbol{\eta}_0(\boldsymbol{\theta}) + \boldsymbol{\lambda}_t^*, \boldsymbol{\theta}) \tag{15}$$

This proposed objective is still a lower bound to the marginal likelihood and it corresponds to the ELBO in Eq. (2) with a distribution whose natural parameter is $\hat{\boldsymbol{\eta}}_t(\boldsymbol{\theta}) = \boldsymbol{\eta}_0(\boldsymbol{\theta}) + \boldsymbol{\lambda}_t^*$ and thus depends on $\boldsymbol{\theta}$. We denote this distribution by $q_{\mathbf{f}}(\mathbf{f}; \hat{\boldsymbol{\eta}}_t(\boldsymbol{\theta}))$. The ELBO is different from the one the distribution obtained after the E-step, with natural parameter $\boldsymbol{\eta}_t^*$ which is independent of $\boldsymbol{\theta}$. We denote this distribution by $q_{\mathbf{f}}(\mathbf{f}; \boldsymbol{\eta}_t^*)$. Clearly, at $\boldsymbol{\theta} = \boldsymbol{\theta}_t$ both objectives match, and so do their gradient with respect to $\boldsymbol{\theta}$, but they generally differ otherwise. We argue that the proposed M-step could lead to a tighter lower bound; see Fig. 1 for an illustration.

In the standard M-step, the dependency of the bound on $\boldsymbol{\theta}$ is only via the KL divergence in Eq. (2). In the M-step we propose, this dependency is more intricate because the expected log-likelihood also depend on $\boldsymbol{\theta}$. Yet, as we show now, it remains simple to implement. The lower bound in the proposed M-step takes a form where an existing implementation of GP regression case can be reused.

$$\mathcal{L}_\eta(\boldsymbol{\eta}_0(\boldsymbol{\theta}) + \boldsymbol{\lambda}_t^*, \boldsymbol{\theta}) = \mathbb{E}_{q_{\mathbf{f}}(\mathbf{f}; \hat{\boldsymbol{\eta}}_t(\boldsymbol{\theta}))} \left[ \log \frac{\prod_{i=1}^n p(y_i \,|\, f_i) \cancel{p_{\boldsymbol{\theta}}(\mathbf{f})}}{\frac{1}{\mathcal{Z}_t(\boldsymbol{\theta})} \prod_{i=1}^n t_i^*(f_i) \cancel{p_{\boldsymbol{\theta}}(\mathbf{f})}} \right] = \log \mathcal{Z}_t(\boldsymbol{\theta}) + c(\boldsymbol{\theta}), \tag{16}$$

where $c(\boldsymbol{\theta}) = \sum_{i=1}^n \mathbb{E}_{q_{\mathbf{f}}(\mathbf{f}; \hat{\boldsymbol{\eta}}_t(\boldsymbol{\theta}))} \left[ \log \frac{p(y_i \,|\, f_i)}{t_i^*(f_i)} \right]$ and $\log \mathcal{Z}_t(\boldsymbol{\theta})$ is the log-partition of $q_{\mathbf{f}}(\mathbf{f}; \hat{\boldsymbol{\eta}}_t(\boldsymbol{\theta}))$,

$$\log \mathcal{Z}_t(\boldsymbol{\theta}) = -\tfrac{n}{2} \log(2\pi) - \tfrac{1}{2} \log |\operatorname{diag}(\boldsymbol{\beta}_t^*)^{-1} + \mathbf{K}_{\mathbf{ff}}(\boldsymbol{\theta})| - \tfrac{1}{2} \widetilde{\mathbf{y}}^\top \left[ \operatorname{diag}(\boldsymbol{\beta}_t^*)^{-1} + \mathbf{K}_{\mathbf{ff}}(\boldsymbol{\theta}) \right]^{-1} \widetilde{\mathbf{y}}. \tag{17}$$

Here, $\widetilde{\mathbf{y}}$ is a vector of $\widetilde{y}_i = -\tfrac{1}{2} \lambda_{1,i}^* / \lambda_{2,i}^*$, and we have explicitly written $\mathbf{K}_{\mathbf{ff}}(\boldsymbol{\theta})$ to show its direct dependence on the hyperparameter $\boldsymbol{\theta}$. The gradients of $\mathcal{Z}_t(\boldsymbol{\theta})$ can be obtained using GP regresssion code, while the gradient of $c(\boldsymbol{\theta})$ can be obtained using standard Monte-Carlo methods. A similar lower bound was originally used in the implementation[2] provided by Khan and Lin [17], but they did not use it for hyperparameter learning.

For GP regression, we recover the exact log-marginal likelihood $\log p_{\boldsymbol{\theta}}(\mathbf{y} \,|\, \mathcal{D})$ for all values of $\boldsymbol{\theta}$. Indeed $\boldsymbol{\lambda}_i^* = (y_i/\sigma^2, -1/(2\sigma^2))$, which means that the sites exactly match the likelihood terms so $c(\boldsymbol{\theta}) = 0$. This also gives us $\widetilde{y}_i = y_i$ and $\beta_i = 1/\sigma^2$, and we get $\log \mathcal{Z}_t(\boldsymbol{\theta}) = \log p_{\boldsymbol{\theta}}(\mathbf{y} \,|\, \mathcal{D})$.

For non-conjugate problems, we found it to be tighter bound than the standard ELBO (Eq. (14))) which could speed-up the procedure. This is illustrated in Fig. 2 (top row) where the proposed ELBO is compared to two other parameterizations (mean-covariance and whitened) for many values of $\boldsymbol{\theta}_t = \boldsymbol{\theta}_{\text{old}}$. We see that the maximum value (shown with a dot) remains rather stable for the proposed method compared to the other two. This is as expected due to Eq. (15) where we expect the solutions to become less sensitive to $\boldsymbol{\theta}_t$ because we have replaced $\boldsymbol{\eta}_0(\boldsymbol{\theta}_t)$ by $\boldsymbol{\eta}_0(\boldsymbol{\theta})$. The bottom row in Fig. 2 shows the iterative steps $(\boldsymbol{\theta}_{t+1}, \boldsymbol{\theta}_t)$ for a few iterations, where we see that, due to the stable solutions of the new ELBO, the iterations quickly converge to the optimum. Exact theoretical reasons behind the speed-ups are currently unknown to us. We believe that the conditioning of the ELBO is improved under the new parameterization. We provide some conditions in App. A under which the new ELBO would provably be tighter.

## 3.2 Faster Natural Gradients for Inference Using the Dual Paramterization

So far, we have assumed that the both E and M steps are run until convergence, but it is more practical to use a stochastic procedure with partial E and M steps, for example, such as those used in [10, 12]. Fortunately, with natural-gradient descent, we can ensure that the iterations also follow the same structure as that of the solution shown in Eq. (9) and Eq. (11). Specifically, we use the method of

---

[2]See `https://github.com/emtiyaz/cvi/blob/master/gp/infKL_cvi.m`

Khan and Lin [17], expressed in terms of the dual parameters $\boldsymbol{\lambda}$ and natural gradients of the expected log-likelihoods $\mathbf{g}_i^{(k)} = \nabla_{\boldsymbol{\mu}_i}\mathbb{E}_{q_{\mathbf{f}}^{(k)}(f_i)}[\log p(y_i \mid f_i)]$ (see also [19, Sec. 5.4]),

$$q_{\mathbf{f}}^{(k+1)}(\mathbf{f}) \propto p_{\boldsymbol{\theta}}(\mathbf{f}) \prod_{i=1}^{n} \underbrace{e^{\langle \boldsymbol{\lambda}_i^{(k+1)}, \mathbf{T}(f_i)\rangle}}_{t_i^{(k+1)}(f_i)}, \text{ where } \boldsymbol{\lambda}_i^{(k+1)} = (1-r_k)\boldsymbol{\lambda}_i^{(k)} + r_k \mathbf{g}_i^{(k)}. \qquad (18)$$

The convergence of these iterations is guaranteed under mild conditions discussed in [21]. The natural parameter of $q_{\mathbf{f}}^{(k)}(\mathbf{f})$ at iteration $k$ can be written in terms of $\boldsymbol{\theta}$ as follows,

$$\boldsymbol{\eta}^{(k)} = \boldsymbol{\eta}_0(\boldsymbol{\theta}) + \boldsymbol{\lambda}^{(k)}, \qquad (19)$$

and the expectation parameters $\boldsymbol{\mu}_i^{(k)}$, required to compute the natural gradients of the expected log-likelihood, can be obtained by using a map from the natural parameter $\boldsymbol{\eta}^{(k)}$.

The updates hold for any $\boldsymbol{\theta}$ and can be conveniently used as $\boldsymbol{\theta} = \boldsymbol{\theta}_t$ at the E-step of the $t^{\text{th}}$ EM iteration. We name $t$-VGP the EM-like algorithm with 1) an E-step consisting of the natural gradient updates of Eq. (18), and, 2) the proposed M-step introduced in Eq. (15).

## 4 The $t$-SVGP Method: Dual-Parameter Based Inference for SVGP

We now extend the dual parameterization based stochastic VEM procedure to the SVGP case and refer to the resulting algorithm as $t$-SVGP. The optimality property shown in Eq. (9) is shared by the ELBO given in Eq. (8). That is, we can express the optimal $q_{\mathbf{u}}^*(\mathbf{u})$ in terms of $n$ 2D parameters $\boldsymbol{\lambda}_i^*$,

$$q_{\mathbf{u}}^*(\mathbf{u}) \propto p_{\boldsymbol{\theta}}(\mathbf{u}) \prod_{i=1}^{n} \underbrace{e^{\langle \boldsymbol{\lambda}_i^*, \mathbf{T}(\mathbf{a}_i^\top \mathbf{u})\rangle}}_{t_i^*(\mathbf{u})}, \text{ where } \boldsymbol{\lambda}_i^* = \nabla_{\boldsymbol{\mu}_{\mathbf{u},i}}\mathbb{E}_{q_{\mathbf{u}}^*(f_i)}[\log p(y_i \mid f_i)]. \qquad (20)$$

The difference here is that the site parameters use the sufficient statistics $\mathbf{T}(\mathbf{a}_i^\top \mathbf{u})$, defined via the projections $\mathbf{a}_i = \mathbf{K}_{\mathbf{uu}}^{-1}\mathbf{k}_{\mathbf{u}i}$. The natural gradients of the expected log-likelihood are computed by using the marginal $q_{\mathbf{u}}^*(f_i)$ defined in Eq. (6) by using $\boldsymbol{\xi}_{\mathbf{u}}^* = (\boldsymbol{\mu}_{\mathbf{u}}^*, \mathbf{S}_{\mathbf{u}}^*)$ evaluated at the expectation parameters $\boldsymbol{\mu}_{\mathbf{u},i}^*$. Note that both $\mathbf{a}_i$ and $q_{\mathbf{u}}^*(f_i)$ depend on $\boldsymbol{\theta}$, but we have suppressed the subscript for notation simplicity.

Similarly to the VGP case, the $\boldsymbol{\lambda}_i^*$ are the optimal dual parameters that measure the sensitivity of the solution to the perturbation in the moments of the posterior marginal $q_{\mathbf{u}}^*(f_i)$. This suggests that we can design a similar VEM procedure that exploits the structure of solution in Eq. (20). The structure is shown below in terms of the natural parameterization of $q_{\mathbf{u}}^*(\mathbf{u})$ for sufficient statistics $\mathbf{T}(\mathbf{u})$,

$$(\mathbf{S}_{\mathbf{u}}^*)^{-1}\mathbf{m}_{\mathbf{u}}^* = \mathbf{K}_{\mathbf{uu}}^{-1}\underbrace{\left(\sum_{i=1}^{n}\mathbf{k}_{\mathbf{u}i}\lambda_{1,i}^*\right)}_{=\bar{\boldsymbol{\lambda}}_1^*} \text{ and } (\mathbf{S}_{\mathbf{u}}^*)^{-1} = \mathbf{K}_{\mathbf{uu}}^{-1} + \mathbf{K}_{\mathbf{uu}}^{-1}\underbrace{\left(\sum_{i=1}^{n}\mathbf{k}_{\mathbf{u}i}\lambda_{2,i}^*\mathbf{k}_{\mathbf{u},i}^\top\right)}_{=\bar{\boldsymbol{\Lambda}}_2^*}\mathbf{K}_{\mathbf{uu}}^{-1}, \quad (21)$$

the quantities $\mathbf{K}_{\mathbf{uu}}$ and $\mathbf{k}_{\mathbf{u}i}$ directly depend on $\boldsymbol{\theta}$ and we can express the ELBO as the partition function of a Gaussian distribution, similarly to Eq. (17) (exact expression in App. B). For large data sets, storing all the $\{\boldsymbol{\lambda}_i^*\}_{i=1}^{n}$ might be problematic, and we can instead store only $\bar{\boldsymbol{\lambda}}_1^*$, a $m$-length vector, and $\bar{\boldsymbol{\Lambda}}_2^*$, a $m \times m$ matrix. This *tied* parameterization is motivated from the *site-tying* setting in sparse EP [1, 24] where the goal is to reduce the storage. The parameterization ignores the dependency of $\mathbf{k}_{\mathbf{u}i}$ over $\boldsymbol{\theta}$ and may reduce the coupling between $\boldsymbol{\theta}$ and $q_{\mathbf{u}}^*$, but it is suitable for large data sets. An alternative tying method consists in storing the sums and the flanking $\mathbf{K}_{\mathbf{uu}}^{-1}$ terms.

We detail the final stochastic variational procedure which we refer to as $t$-SVGP: Given a parameter $\boldsymbol{\theta}_t$, we run a few iterations of the E-step. At each iteration $k$ of the E-step, given $\mathbf{m}_{\mathbf{u}}^{(k)}$ and $\mathbf{S}_{\mathbf{u}}^{(k)}$, we sample a minibatch $\mathcal{M}$ and compute the natural gradients by first computing

$$\alpha_i^{(k)} = \mathbb{E}_{q_{\mathbf{u}}^{(k)}(f_i)}\left[\nabla_f \log p(y_i \mid f_i)\right] \quad \text{ and } \quad \beta_i^{(k)} = \mathbb{E}_{q_{\mathbf{u}}^{(k)}(f_i)}\left[-\nabla_{ff}^2 \log p(y_i \mid f_i)\right],$$

using the marginals $q_{\mathbf{u}}^{(k)}(f_i)$ from Eq. (6). The natural gradients of the expected log-likelihood for the $i^{\text{th}}$ site is then equal to $\mathbf{g}_i^{(k)} = (\beta_i^{(k)}m_i^{(k)} + \alpha_i^{(k)}, \ \beta_i^{(k)})$. Using these natural gradients we can use an iterative procedure similar to Eq. (18) but now on the tied parameters,

$$\bar{\boldsymbol{\lambda}}_1^{(k+1)} \leftarrow (1-r_k)\bar{\boldsymbol{\lambda}}_1^{(k)} + r_k \sum_{i\in\mathcal{M}} \mathbf{k}_{\mathbf{u}i}\mathbf{g}_{1,i}^{(k)}, \qquad (22)$$

$$\bar{\boldsymbol{\Lambda}}_2^{(k+1)} \leftarrow (1-r_k)\bar{\boldsymbol{\Lambda}}_2^{(k)} + r_k \sum_{i\in\mathcal{M}} \mathbf{k}_{\mathbf{u}i}\mathbf{k}_{\mathbf{u}i}^\top \mathbf{g}_{2,i}^{(k)}. \qquad (23)$$

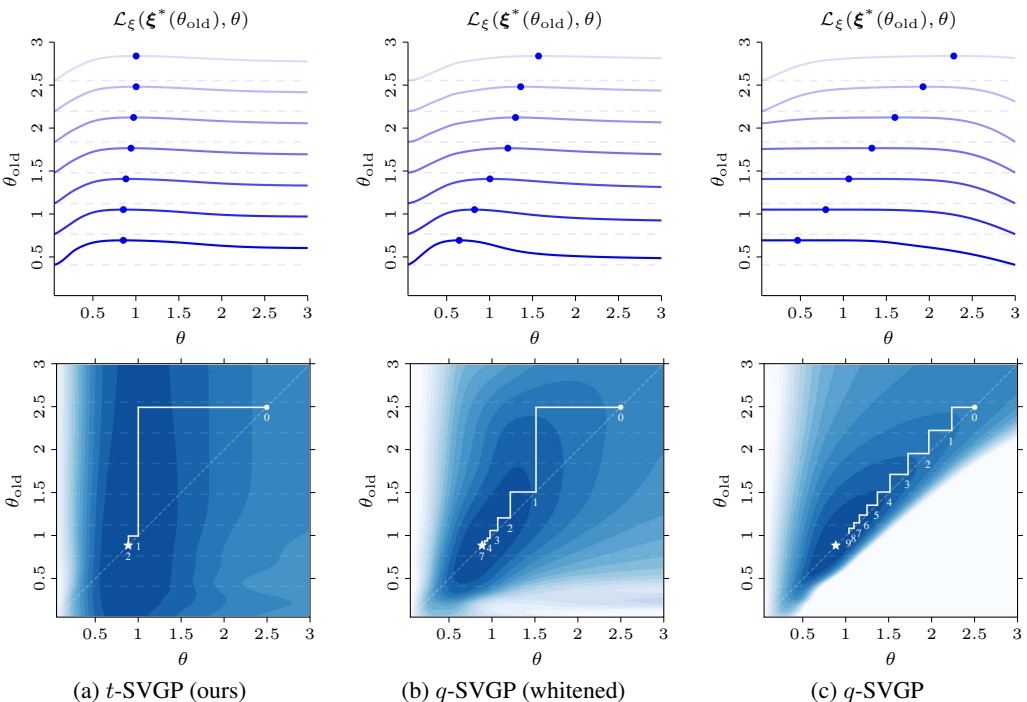

Figure 2: EM iterations for hyperparameter learning on a classification task on a a toy dataset. **Top row:** for the M-step, the optima of the $t$-SVGP objective (left) are less sensitive to the initial hyperparameter value $\boldsymbol{\theta}_{\text{old}}$, compared to $q$-SVGP (middle and right). **Bottom row:** the EM iterations are shown in white on top of the EM objectives for a range of starting values for $\boldsymbol{\theta}_{\text{old}}$. The starting point $\theta = 2.5$ is marked with a white dot and the optimum with a star. $t$-SVGP converges much faster (2 iterations in leftmost plot) compared to $q$-SVGP (middle and rightmost plots which take $>5$ iterations).

The natural parameter required can be obtained using Eq. (21),

$$\mathbf{S}_{\mathbf{u}}^{(k)} \leftarrow \left(\mathbf{K}_{\mathbf{uu}}^{-1} + \mathbf{K}_{\mathbf{uu}}^{-1}\bar{\boldsymbol{\Lambda}}_2^{(k)}\mathbf{K}_{\mathbf{uu}}^{-1}\right)^{-1} \quad \text{and} \quad \mathbf{m}_{\mathbf{u}}^{(k)} \leftarrow \mathbf{S}_{\mathbf{u}}^{(k)}\mathbf{K}_{\mathbf{uu}}^{-1}\bar{\boldsymbol{\lambda}}_1^{(k)}. \tag{24}$$

After a few E-steps, we update the parameters with a gradient descent step using the gradient of the log-partition function of $q_{\mathbf{f}}^{(k)}(\mathbf{u})$ with respect to $\boldsymbol{\theta}$. In App. C, we detail how to efficiently make predictions and compute the ELBO under parameterization Eq. (21). The full algorithm is given in App. E. The convergence of the sequence of stochastic updates in the E-step is guaranteed under mild conditions discussed in [21] for the untied setting. Site-tying introduces a bias but does not seem to affect convergence in practice.

## 5  Empirical Evaluation

We conduct experiments to highlight the advantages of using the dual parameterization. Firstly, we study the effects of the improved objective for hyperparameter learning of $t$-SVGP versus $q$-SVGP. We study the objective being optimized for a single M-step, after an E-step ran until convergence. We then show a full sequence of EM iterations on small data sets. For large-scale data, where running steps to convergence is expensive, we use partial E and M-steps and mini-batching. Our improved bound and faster natural gradient computations show benefits in both settings. It is worth noting that the E-step for both $q$-SVGP with natural gradients and $t$-SVGP are identical up to machine precision, and any differences in performance are to be attributed to the different parameterization.

**The Role of the Learning Objective** In Fig. 1, we learn the kernel hyperparameters $\boldsymbol{\theta}$ in a GP classification task via coordinate ascent of the lower bound $\mathcal{L}_{\xi}(\boldsymbol{\xi}, \boldsymbol{\theta})$, where $\boldsymbol{\xi}$ are the variational parameters, *i.e.* via EM. Starting at hyperparameter $\boldsymbol{\theta}_{\text{old}}$, we denote by $\boldsymbol{\xi}^*(\boldsymbol{\theta}_{\text{old}})$ the associated optimal

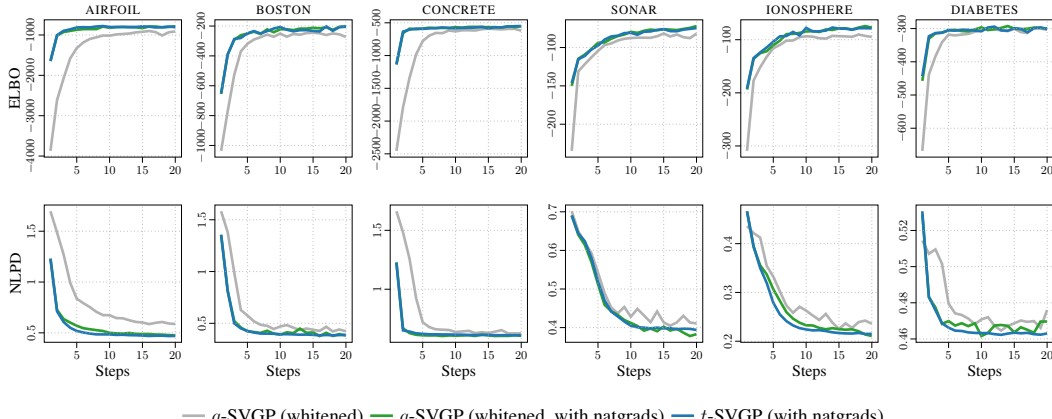

Figure 3: Comparison of convergence in terms of ELBO and negative log-predictive density (NLPD) as averages over 5-fold cross-validation runs, on UCI regression and classification tasks. All methods were trained (incl. hyperparameters and 50 inducing input locations) with matching learning rate. Natural gradient–based training is superior, and the $t$-SVGP parameterization improves stability.

variational parameters. Updating $\theta$ consists in optimizing $\mathcal{L}_\xi(\xi^*(\theta_{\text{old}}), \theta)$ which we show on the left panel for the dual parameterization (blue), the standard whitened (orange) and unwhitened (green) SVGP parameterizations. The dual parameterization leads to a tighter bound and thus to bigger steps and faster overall convergence as shown on the right for the illustrative toy classification task, starting at $(\theta_1, \theta_2) = (1, 1)$, in the extreme case of taking both the E and M step to convergence. For the toy data set we use $m = 10$ inducing points (see details in App. F).

In Fig. 2, we also use the toy data and parameters as in Fig. 1, but we show how the learning objective changes over iterations. The blue contours show, for all initial $\theta_{\text{old}}$, the objective maximized in the M-step, *i.e.* $\mathcal{L}_\xi(\xi^*(\theta_{\text{old}}), \theta)$. The orange lines show, for all initial $\theta_{\text{old}}$, the outcome of an E-step followed by and M-step, *i.e.* $\theta^*(\theta_{\text{old}}) = \arg\max_\theta \mathcal{L}_\xi(\xi^*(\theta_{\text{old}}), \theta)$. The EM iterations converge to the fixed points of $\theta^*$, *i.e.* its intersection with the diagonal line of the identity function. A flatter line around the optimal value is more desirable as it means the iterations converge faster to the optimum value which in this experiment is just below one, while a line close to the diagonal leads to slow convergence. Here $t$-SVGP has the fastest convergence, $q$-SVGP performs poorly, although whitening clearly helps the optimisation problem. The dark dashed lines show how optimising $\theta$ would look starting from $\theta_0 = 2.5$ and running 8 iterations for the different models.

**Evaluation on UCI Classification and Regression Tasks** We use common small and mid-sized UCI data sets to test the performance of our method against $q$-SVGP with natural gradient optimisation and normal $q$-SVGP trained with Adam optimizer for the variational parameters. All methods use Adam for the hyperparameters. The exact details of the data sets can be found in App. F. Here we again take the approach that the optimal way to optimize the ELBO if computational budget allows is to alternate between performing E and M-steps till convergence. We plot how the different inference schemes perform for ELBO and NLPD on a hold test set. We perform 5-fold cross validation with the results in Fig. 3 showing the mean of the folds for ELBO and NLPD. Natural gradient variants of $q$-SVGP clearly perform better than non natural gradient $q$-SVGP. Our method $t$-SVGP seems more stable specifically in NLPD for most data sets if not equal to $q$-SVGP. For $q$-SVGP, we have used the whitened version which as we noted helps with hyperparameter optimisation.

**Improved Efficiency in Large-scale Inference** To highlight *practical* benefits, we show the performance of our stochastic and sparse $t$-SVGP framework on the MNIST ([23], available under CC BY-SA 3.0) multiclass-classification task (for details see App. F). Given the data set is $n = 70,000$, minibatching is needed and we adopt the parameterization of Eq. (21) meaning we match $q$-SVGP for parameter storage complexity. We compare against the natural gradient $q$-SVGP implementation but not the non-natural gradient version since it produces considerably worse performance. In large scale minibatching experiments performing full E and M steps may not be efficient. Instead we perform partial steps for both. A single E and M step can be thought of as the approach outlined in [33]. All experiments are performed with a batch size of $n_{\text{b}} = 200$ and $m = 100$ inducing points and the

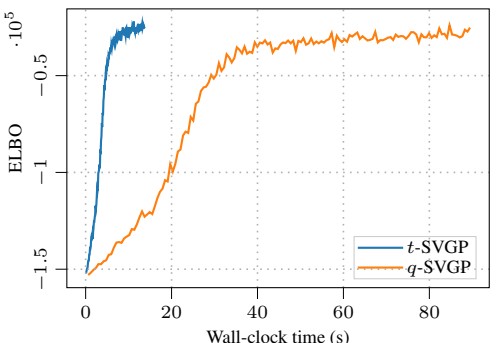 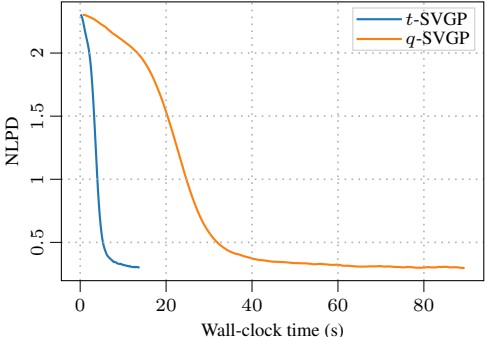

Figure 4: Comparison of practical inference and learning on MNIST. We compare training time on a laptop between $t$-SVGP to the $q$-SVGP model in GPflow in terms of wall-clock time of training for 150 steps, where both methods use natural gradient updates and share the same learning rates.

optimization is ran until convergence using the Adam optimizer for the hyperparameters (M-step). Table 1 shows different variations of learning rates and iterations of E- and M-steps. The results suggest some benefits in running partial EM steps. The $t$-SVGP formulation performs equally if not better than $q$-SVGP under all settings.

In Fig. 4, we show the speed advantage of $t$-SVGP over $q$-SVGP due to cheaper natural gradient updates. We compare against the state-of-the-art implementation of SVGP in GPflow ([26], v2.2.1) and a closely matched implementation of our method in GPflow. We compare wall-clock time to compute 150 steps of the algorithm for both methods in terms of NLPD and ELBO taking single E and M-steps (MacBook pro, 2 GHz CPU,

Table 1: NLPD on MNIST benchmarks for different learning rates and E and M steps.

| NLPD | | LR | | STEPS | |
| $q$-SVGP | $t$-SVGP | E | M | #E | #M |
| --- | --- | --- | --- | --- | --- |
| **0.304**±**0.015** | **0.304**±**0.006** | 0.040 | 0.05 | 1 | 1 |
| 0.289±0.010 | **0.283**±**0.007** | 0.035 | 0.10 | 2 | 1 |
| 0.293±0.020 | **0.281**±**0.010** | 0.030 | 0.10 | 3 | 1 |
| 0.259±0.010 | **0.255**±**0.006** | 0.025 | 0.03 | 4 | 2 |
| **0.282**±**0.007** | **0.283**±**0.006** | 0.050 | 0.03 | 4 | 2 |
| 0.243±0.003 | **0.230**±**0.009** | 0.030 | 0.03 | 4 | 1 |

16 GB RAM). Our implementation avoids the use of sluggish automatic differentiation to compute the natural gradients, and, even if our implementation is not as optimized as SVGP in GPflow, it is roughly 5 times faster on this standard benchmark.

## 6 Discussion and Conclusion

Sparse variational GP (SVGP) methods are the current *de facto* approach to allow GPs to scale to large problems. In this paper, we introduced an alternative parameterization to variational GPs that leads to an improved loss landscape for learning (*cf.*, Fig. 1). This improvement hinges on writing the variational problem in terms of its dual—similar to the conjugate-computation variational inference (CVI) approach by Khan and Lin [17]—parameterization to capture sites: we assume the approximate posterior decomposes into a prior contribution and a Gaussian *approximate* likelihood contribution. Variational inference under this model can conveniently be implemented by mirror descent and corresponds to natural gradient based learning, thus improving convergence in variational parameter optimization (the 'E-step'), at the same time as improving hyperparameter optimization (the 'M-step'), due to the tighter evidence lower bound.

We further show that we can derive the *sparse* equivalent of this method, which also allows for stochastic training through mini-batching, reducing the computational complexity to $\mathcal{O}(m^3 + n_{\mathrm{b}}m^2)$ per step. Our method matches the asymptotic computational cost of other SVGP methods, while marginally reducing compute due to simpler expressions to back-propagate through (see discussion in Sec. 5). Our empirical validation across a wide variety of regression and classification tasks confirms the benefits suggested by our theory: The proposed strategy typically allows for improved stability over gold-standard SVGP methods even when the learning rates remain the same. It allows for higher learning rates, and reduces computational cost—leading to improved learning both in terms of reduced steps as well as expected wall-clock time.

We provide a reference implementation of our method under the GPflow framework at `https://github.com/AaltoML/t-SVGP`.

## Acknowledgments and Disclosure of Funding

AS acknowledges funding from the Academy of Finland (grant numbers 324345 and 339730). We acknowledge the computational resources provided by the Aalto Science-IT project. We thank Stefanos Eleftheriadis, Richard E. Turner, and Hugh Salimbeni for comments on the manuscript.

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
