# Supplementary Material for
# Dual Parameterization of Sparse Variational Gaussian Processes

## A  Tighter Bound for the M-step

We here study the role of parameterizations $\boldsymbol{\xi}$ in shaping the losses optimized during the M-step of the EM learning procedure. Each parameterization $\boldsymbol{\xi}$ has associated natural parameters $\boldsymbol{\eta}$.

We introduce an alternative expression of the loss $\mathcal{L}$ in terms of the natural parameters of the prior $\boldsymbol{\eta}_p$ and of the approximate posterior $\boldsymbol{\eta}_q$: $\mathcal{L}(q, \boldsymbol{\theta}) = L(\boldsymbol{\eta}_q, \boldsymbol{\eta}_p)$. To simplify the presentation but without loss of generality, we consider the case $\boldsymbol{\theta} = \boldsymbol{\eta}_p$, i.e. when the hyperparameters are directly the natural parameters. The case we actually care about is when $\boldsymbol{\theta}$ indexes natural parameters $\boldsymbol{\eta}_p(\boldsymbol{\theta})$, in which case, the natural parameters lie on a manifold in $\Omega$.

We focus on the difference between parameterizations where the posterior statistics $\boldsymbol{\eta}_q$ depends on the prior statistics $\boldsymbol{\eta}_p$, as in the dual parameterization $\boldsymbol{\lambda}$, where this dependence is linear $\boldsymbol{\eta}_q = \boldsymbol{\eta}_p + \boldsymbol{\lambda}$, versus parameterizations that don't, as in the $\boldsymbol{\xi} = (\boldsymbol{\mu}, \mathbf{L})$ parameterization. To make this distinction explicit we introduce the losses

$$\tilde{l}(\boldsymbol{\eta}_p) = L(\boldsymbol{\eta}_p + \boldsymbol{\lambda}^*, \boldsymbol{\eta}_p), \tag{25}$$

$$l(\boldsymbol{\eta}_p) = L(\boldsymbol{\eta}_q^*, \boldsymbol{\eta}_p). \tag{26}$$

For a matched optimal E-step, *i.e.* $\boldsymbol{\eta}_p + \boldsymbol{\lambda}^* = \boldsymbol{\eta}_q^* = \arg\max_{\boldsymbol{\eta}} L(\boldsymbol{\eta}, \boldsymbol{\eta}_p)$, the value of $l$ and $\tilde{l}$ and their gradient w.r.t. $\boldsymbol{\eta}_p$ are the same:

$$\tilde{l}(\boldsymbol{\eta}_p) = l(\boldsymbol{\eta}_p), \tag{27}$$

$$\nabla_{\boldsymbol{\eta}_p} \tilde{l}(\boldsymbol{\eta}_p) = \underbrace{\partial_{\boldsymbol{\eta}_1} L|_{\boldsymbol{\eta}_{q^*}}}_{=0} + \partial_{\boldsymbol{\eta}_2} L|_{\boldsymbol{\eta}_p} = \partial_{\boldsymbol{\eta}_2} \mathcal{L}|_{\boldsymbol{\eta}_p} = \nabla_{\boldsymbol{\eta}_p} l(\boldsymbol{\eta}_p). \tag{28}$$

In the conjugate regression case, we have that $\tilde{l}(\boldsymbol{\eta}_p) \geq l(\boldsymbol{\eta}_p)$:

$$\tilde{l}(\boldsymbol{\eta}_p) - l(\boldsymbol{\eta}_p) = -(\log p(\mathcal{D}) - \tilde{l}(\boldsymbol{\eta}_p)) + (\log p(\mathcal{D}) - l(\boldsymbol{\eta}_p)) \tag{29}$$

$$= -\underbrace{\mathrm{D}_{\mathrm{KL}}[\boldsymbol{\eta}_p + \boldsymbol{\lambda}^* \,\|\, \boldsymbol{\eta}_{\mathrm{post}}]}_{=0} + \mathrm{D}_{\mathrm{KL}}[\boldsymbol{\eta}_{q^*} \,\|\, \boldsymbol{\eta}_{\mathrm{post}}] \tag{30}$$

$$= \mathrm{D}_{\mathrm{KL}}[\boldsymbol{\eta}_{q^*} \,\|\, \boldsymbol{\eta}_{\mathrm{post}}] > 0. \tag{31}$$

We can't show this in the non-conjugate setting but instead focus on the local behavior of $\tilde{l}(\boldsymbol{\eta}_p)$ and $l(\boldsymbol{\eta}_p)$. Specifically, since their gradients match, we study their Hessians, which are different:

$$\nabla^2_{\boldsymbol{\eta}_p \boldsymbol{\eta}_p} \tilde{l}(\boldsymbol{\eta}_p) = \partial^2_{\boldsymbol{\eta}_1 \boldsymbol{\eta}_1} L|_{\boldsymbol{\eta}_{q^*}} + \partial^2_{\boldsymbol{\eta}_2 \boldsymbol{\eta}_2} L|_{\boldsymbol{\eta}_p} + 2\partial^2_{\boldsymbol{\eta}_1 \boldsymbol{\eta}_2} L|_{\boldsymbol{\eta}_{q^*} \boldsymbol{\eta}_p} \tag{32}$$

$$\nabla^2_{\boldsymbol{\eta}_p \boldsymbol{\eta}_p} l(\boldsymbol{\eta}_p) = \partial^2_{\boldsymbol{\eta}_2 \boldsymbol{\eta}_2} L|_{\boldsymbol{\eta}_p} \tag{33}$$

The Hessian difference between the two conditions is

$$\Delta H = \partial^2_{\boldsymbol{\eta}_1 \boldsymbol{\eta}_1} L|_{\boldsymbol{\eta}_{q^*}} + 2\partial^2_{\boldsymbol{\eta}_1 \boldsymbol{\eta}_2} L|_{\boldsymbol{\eta}_{q^*} \boldsymbol{\eta}_p} \tag{34}$$

and using the identity

$$\partial^2_{\boldsymbol{\eta}_1 \boldsymbol{\eta}_2} L|_{\boldsymbol{\eta}_{q^*} \boldsymbol{\eta}_p} = -\partial^2_{\boldsymbol{\eta}_1 \boldsymbol{\eta}_2} D_{KL}(\boldsymbol{\eta}_t + \boldsymbol{\eta}_p, \boldsymbol{\eta}_p)|_{\boldsymbol{\eta}_{q^*} \boldsymbol{\eta}_p} = \mathbf{I}[\boldsymbol{\eta}_{q^*}]. \tag{35}$$

The Hessian difference can be expressed as

$$\Delta H = \partial^2_{\boldsymbol{\eta}_1 \boldsymbol{\eta}_1} L|_{\boldsymbol{\eta}_{q^*}} + 2\mathbf{I}[\boldsymbol{\eta}_{q^*}]. \tag{36}$$

$\tilde{l}(\boldsymbol{\eta}_p)$ is a local upper bound to $l(\boldsymbol{\eta}_p)$ if $\Delta H \succeq 0$

$$\Delta H \succeq 0 \iff \partial^2_{\boldsymbol{\eta}_1 \boldsymbol{\eta}_1} L|_{\boldsymbol{\eta}_{q^*}} \succeq -2\mathbf{I}[\boldsymbol{\eta}_{q^*}]. \tag{37}$$

This corresponds to a condition on the curvature of the optimization problem in the preceding E-step. We can verify that this condition is met in the conjugate case where

$$\partial^2_{\boldsymbol{\eta}_1 \boldsymbol{\eta}_1} L|_{\boldsymbol{\eta}_{q^*}} = -\partial^2_{\boldsymbol{\eta}_1 \boldsymbol{\eta}_1} D_{\mathrm{KL}}[\boldsymbol{\eta}_{q^*} \| \boldsymbol{\eta}_{\mathrm{post}}] = -\mathbf{I}[\boldsymbol{\eta}_{q^*}]. \tag{38}$$

The condition is indeed met since the Fisher information matrix $\mathbf{I}[\boldsymbol{\eta}_{q^*}]$ is positive semi-definite.

## B  Proposed Objective for the M-step of $t$-SVGP

Starting from hyperparameter $\boldsymbol{\theta}_{\mathrm{old}}$, an E-step gives the optimal dual parameters $\boldsymbol{\lambda}^*$. The objective for the proposed M-step of $t$-SVGP is the ELBO in Eq. (8) for the variational distribution $q_{\mathbf{u}}(\mathbf{u}; \hat{\boldsymbol{\eta}}_{\mathbf{u}}(\boldsymbol{\theta}))$ with $\boldsymbol{\theta}$ dependent parameters $\hat{\boldsymbol{\eta}}_{\mathbf{u}}(\boldsymbol{\theta})$ expressed in terms of the mean and covariance matrix as

$$\hat{\mathbf{S}}_{\mathbf{u}}^{-1}\hat{\mathbf{m}}_{\mathbf{u}} = \mathbf{K}_{\mathbf{uu}}^{-1} \underbrace{\left( \sum_{i=1}^n \mathbf{k}_{\mathbf{u}i} \lambda_{1,i}^* \right)}_{=\bar{\boldsymbol{\lambda}}_1} \text{ and } \hat{\mathbf{S}}_{\mathbf{u}}^{-1} = \mathbf{K}_{\mathbf{uu}}^{-1} + \mathbf{K}_{\mathbf{uu}}^{-1} \underbrace{\left( \sum_{i=1}^n \mathbf{k}_{\mathbf{u}i} \lambda_{2,i}^* \mathbf{k}_{\mathbf{u},i}^\top \right)}_{=\bar{\boldsymbol{\Lambda}}_2} \mathbf{K}_{\mathbf{uu}}^{-1}. \tag{39}$$

Introducing $q(\mathbf{f}, \mathbf{u}; \boldsymbol{\theta}) = p_{\boldsymbol{\theta}}(\mathbf{f}|\mathbf{u}) q_{\mathbf{u}}(\mathbf{u}; \hat{\boldsymbol{\eta}}_{\mathbf{u}}(\boldsymbol{\theta}))$, the ELBO for our proposed M-step is given by:

$$\begin{aligned} \mathcal{L}_{\eta_u}(\hat{\boldsymbol{\eta}}_{\mathbf{u}}(\boldsymbol{\theta}), \boldsymbol{\theta}) &= \mathbb{E}_{q(\mathbf{f},\mathbf{u};\boldsymbol{\theta})} \left[ \log \frac{p_{\boldsymbol{\theta}}(\mathbf{y}, \mathbf{f}, \mathbf{u})}{\hat{q}_t(\mathbf{f}, \mathbf{u}; \boldsymbol{\theta})} \right] \\ &= \mathbb{E}_{q(\mathbf{f},\mathbf{u};\boldsymbol{\theta})} \left[ \log \frac{\prod_{i=1}^n p(y_i \mid f_i) \cancel{p_{\boldsymbol{\theta}}(\mathbf{f}|\mathbf{u})} \cancel{p_{\boldsymbol{\theta}}(\mathbf{u})}}{\frac{1}{\mathcal{Z}(\boldsymbol{\theta})} t^*(\mathbf{u}) \cancel{p_{\boldsymbol{\theta}}(\mathbf{f}|\mathbf{u})} \cancel{p_{\boldsymbol{\theta}}(\mathbf{u})}} \right] \\ &= \log \mathcal{Z}(\boldsymbol{\theta}) + c(\boldsymbol{\theta}), \end{aligned} \tag{40}$$

where $c(\boldsymbol{\theta}) = \sum_{i=1}^n \mathbb{E}_{q_t(f_i;\boldsymbol{\theta})}[\log p(y_i \mid f_i)] - \mathbb{E}_{q_t(\mathbf{u};\boldsymbol{\theta})}[\log t^*(\mathbf{u})]$ and $\log \mathcal{Z}(\boldsymbol{\theta})$ is the log-partition of the Gaussian $q_{\mathbf{u}}(\mathbf{u}; \hat{\boldsymbol{\eta}}_{\mathbf{u}}(\boldsymbol{\theta}))$

$$\begin{aligned} \log \mathcal{Z}(\boldsymbol{\theta}) = -\frac{m}{2} \log(2\pi) - \frac{1}{2} \log |\mathbf{K}_{\mathbf{uu}}(\boldsymbol{\theta}) \bar{\boldsymbol{\Lambda}}_2^{-1} \mathbf{K}_{\mathbf{uu}}(\boldsymbol{\theta}) + \mathbf{K}_{\mathbf{uu}}(\boldsymbol{\theta})| \\ - \frac{1}{2} \widetilde{\mathbf{y}}^\top \left[ \mathbf{K}_{\mathbf{uu}}(\boldsymbol{\theta}) \bar{\boldsymbol{\Lambda}}_2^{-1} \mathbf{K}_{\mathbf{uu}}(\boldsymbol{\theta}) + \mathbf{K}_{\mathbf{uu}}(\boldsymbol{\theta}) \right]^{-1} \widetilde{\mathbf{y}}, \end{aligned} \tag{41}$$

with $\widetilde{\mathbf{y}} = \mathbf{K}_{\mathbf{uu}}(\boldsymbol{\theta}) \bar{\boldsymbol{\Lambda}}_2^{-1} \bar{\boldsymbol{\lambda}}_1$.

## C  Efficient ELBO Computation for $t$-SVGP

We here detail the computations required to perform inference and learning using the dual parameterization. To perform inference, the variational expectations need to be evaluated. These require the evaluation of the marginal predictions $q(f(\mathbf{x}_i))$ for all inputs $\mathbf{x}_i$ in $\mathcal{D}$. For learning, the ELBO in Eq. (8) needs to be evaluated which requires the computation of a KL divergence.

In $t$-SVGP, the variational distribution $q(\mathbf{u}) = \mathrm{N}(\mathbf{u}|\mathbf{m}, \mathbf{S})$ is parameterized in terms of its natural parameters:

$$\mathbf{S}^{-1} = \mathbf{K}_{\mathbf{uu}}^{-1} + \mathbf{K}_{\mathbf{uu}}^{-1} \bar{\boldsymbol{\Lambda}}_2 \mathbf{K}_{\mathbf{uu}}^{-1}, \tag{42}$$

$$\mathbf{S}^{-1}\mathbf{m} = \mathbf{K}_{\mathbf{uu}}^{-1} \bar{\boldsymbol{\lambda}}_1, \tag{43}$$

where

$$\bar{\boldsymbol{\lambda}}_1 = \sum_{i=1}^n \mathbf{k}_{i\mathbf{u}}^\top \lambda_{1,i} \qquad \text{and} \qquad \bar{\boldsymbol{\Lambda}}_2 = \sum_{i=1}^n \mathbf{k}_{i\mathbf{u}}^\top \mathbf{k}_{i\mathbf{u}} \lambda_{2,i}. \tag{44}$$

Introducing $\mathbf{R} = \mathbf{K}_{\mathbf{uu}} + \bar{\boldsymbol{\Lambda}}_2$, the mean and covariance $q(\mathbf{u})$ can be rewritten as:

$$\mathbf{S} = (\mathbf{K}_{\mathbf{uu}}^{-1} + \mathbf{K}_{\mathbf{uu}}^{-1} \bar{\boldsymbol{\Lambda}}_2 \mathbf{K}_{\mathbf{uu}}^{-1})^{-1} \tag{45}$$

$$= \mathbf{K}_{\mathbf{uu}} (\mathbf{K}_{\mathbf{uu}} + \bar{\boldsymbol{\Lambda}}_2)^{-1} \mathbf{K}_{\mathbf{uu}} \tag{46}$$

$$= \mathbf{K}_{\mathbf{uu}} \mathbf{R}^{-1} \mathbf{K}_{\mathbf{uu}}, \tag{47}$$

$$\mathbf{m} = \mathbf{K}_{\mathbf{uu}} \mathbf{R}^{-1} \bar{\boldsymbol{\lambda}}_1. \tag{48}$$

This leads to simple closed form expressions for the marginal predictions:

$$q(\mathbf{f}^\star) = \mathrm{N}(\mathbf{f}^\star | \mathbf{K}_{\star\mathbf{u}}\mathbf{R}^{-1}\bar{\boldsymbol{\lambda}}_1, \mathbf{K}_{\star\star} - \mathbf{K}_{\star\mathbf{u}}\mathbf{K}_{\mathbf{uu}}^{-1}\mathbf{K}_{\mathbf{u}\star} + \mathbf{K}_{\star\mathbf{u}}\mathbf{R}^{-1}\mathbf{K}_{\mathbf{u}\star}), \tag{49}$$

and for the and KL divergence Eq. (8):

$$\mathrm{D}_{\mathrm{KL}}\left(q(\mathbf{u}) \,\|\, p(\mathbf{u})\right) = \tfrac{1}{2}\left(\mathrm{tr}\left(\mathbf{K}_{\mathbf{uu}}^{-1}\mathbf{S}\right) + \mathbf{m}^\top\mathbf{K}_{\mathbf{uu}}^{-1}\mathbf{m} - k + \ln|\mathbf{K}_{\mathbf{uu}}\mathbf{S}^{-1}|\right) \tag{50}$$

$$= \tfrac{1}{2}\left(\mathrm{tr}(\mathbf{K}_{\mathbf{uu}}\mathbf{R}^{-1}) - k + \boldsymbol{\lambda}_1^\top\mathbf{R}^{-1}\mathbf{K}_{\mathbf{uu}}\mathbf{R}^{-1}\bar{\boldsymbol{\lambda}}_1 - \ln|\mathbf{K}_{\mathbf{uu}}| + \ln|\mathbf{R}|\right). \tag{51}$$

# D   Pseudocode for the $q$-SVGP Algorithm

We here detail the $q$-SVGP algorithm for inference and learning with the E-step as described in [2], for parameterization $\boldsymbol{\xi} = (\mathbf{m}, \mathbf{L})$. The pseudocode shows an E-step comprised of $K$ iterations of natural gradient descent, followed by an M-step comprised of $S$ gradient descent iterations with learning rate $\gamma$.

---

**Algorithm 1** $q$-SVGP

---

1: initialization at $\boldsymbol{\theta}_t, \boldsymbol{\xi}_t$
2: **for** $k = 0 \ldots K - 1$ **do**
3:    $\boldsymbol{\xi}^{(0)} \leftarrow \boldsymbol{\xi}_t$                                     *Initialization of the natural gradient descent iterations*
4:    **for** $i = 1 \ldots n$ **do**
5:       $q^{(k)}(f_i) = \int p_{\boldsymbol{\theta}_t}(f_i \,|\, \mathbf{u})q_{\mathbf{u}}^{(k)}(\mathbf{u})\,\mathrm{d}\mathbf{u}$                        *Marginal predictions*
6:    **end for**
7:    $\mathcal{L}^{(k)} = \sum_i \mathbb{E}_{q^{(k)}(f_i)}\left[\log p(y_i \,|\, f_i)\right] - \mathrm{D}_{\mathrm{KL}}\left[q_{\mathbf{u}}^{(k)}(\mathbf{u}) \,\middle\|\, p_{\boldsymbol{\theta}_t}(\mathbf{u})\right]$     *ELBO*
8:    $\boldsymbol{\eta}^{(k)} \leftarrow \boldsymbol{\xi}^{(k)}$                                   *Gaussian transformation*
9:    $\mathbf{g}^{(k)} \leftarrow \nabla_{\boldsymbol{\mu}}\boldsymbol{\xi}(\boldsymbol{\mu}^{(k)})\nabla_{\boldsymbol{\xi}}\mathcal{L}^{(k)}|_{\boldsymbol{\xi}=\boldsymbol{\xi}_t}$                      *Natural gradient*
10:   $\boldsymbol{\eta}^{(k+1)} \leftarrow \boldsymbol{\eta}^{(k)} + \rho\,\mathbf{g}^{(k)}$                           *Natural gradient step*
11:   $\boldsymbol{\xi}^{(k+1)} \leftarrow \boldsymbol{\eta}^{(k+1)}$                            *Gaussian transformation*
12: **end for**
13: $\boldsymbol{\xi}_{t+1} \leftarrow \boldsymbol{\xi}^{(K)}$                                    *End of E-step*
14: $\boldsymbol{\theta}^{(0)} \leftarrow \boldsymbol{\theta}_t$                           *Initialization of the gradient descent iterations*
15: **for** $s = 0 \ldots S - 1$ **do**
16:   $\tilde{\mathcal{L}}^{(s)}(\boldsymbol{\theta}) = -\mathrm{D}_{\mathrm{KL}}\left[q_{\mathbf{u}}^{(s)}(\mathbf{u}) \,\middle\|\, p_{\boldsymbol{\theta}}(\mathbf{u})\right]$                      *KL of ELBO*
17:   $\boldsymbol{\theta}^{(s+1)} \leftarrow \boldsymbol{\theta}^{(s)} + \gamma\nabla_{\boldsymbol{\theta}}\tilde{\mathcal{L}}^{(s)}|_{\boldsymbol{\theta}=\boldsymbol{\theta}^{(s)}}$             *Gradient descent step for $\boldsymbol{\theta}$*
18: **end for**
19: $\boldsymbol{\theta}_{t+1} \leftarrow \boldsymbol{\theta}^{(S)}$                                      *End of M-step*

---

# E   Pseudocode for the $t$-SVGP Algorithm

We here summarize the $t$-SVGP algorithm using the dual parameterization. The pseudocode shows an E-step comprised of $K$ iterations of natural gradient descent, followed by an M-step comprised of $S$ gradient descent iterations with learning rate $\gamma$.

**Algorithm 2** $t$-SVGP

1:  initialization at $\boldsymbol{\theta}_t, \boldsymbol{\lambda}_t$
2:  **for** $k = 0 \dots K - 1$ **do**
3:      $\boldsymbol{\lambda}^{(0)} \leftarrow \boldsymbol{\lambda}_t$                                   *Initialization of the natural gradient descent iterations*
4:      **for** $i = 1 \dots n$ **do**
5:          $q_{\mathbf{u}}^{(k)}(f_i) = \int p_{\boldsymbol{\theta}_t}(f_i \mid \mathbf{u}) q_{\mathbf{u}}^{(k)}(\mathbf{u}; \boldsymbol{\lambda}^{(k)}) \, \mathrm{d}\mathbf{u}$                          *Marginal predictions*
6:          $\alpha_i^{(k)} = \mathbb{E}_{q_{\mathbf{u}}^{(k)}(f_i)}[\nabla_f \log p(y_i \mid f_i)]$
7:          $\beta_i^{(k)} = \mathbb{E}_{q_{\mathbf{u}}^{(k)}(f_i)}\left[-\nabla_{ff}^2 \log p(y_i \mid f_i)\right]$
8:          $\mathbf{g}_i^{(k)} = (\beta_i^{(k)} m_i^{(k)} + \alpha_i^{(k)}, \ \beta_i^{(k)})$                          *Natural gradient*
9:      **end for**
10:     $\bar{\boldsymbol{\lambda}}_1^{(k+1)} \leftarrow (1 - r)\bar{\boldsymbol{\lambda}}_1^{(k)} + r \sum_{i \in \mathcal{M}} \mathbf{k}_{\mathbf{u}i} \mathbf{g}_{1,i}^{(k)}$                          *Natural gradient step*
11:     $\bar{\boldsymbol{\Lambda}}_2^{(k+1)} \leftarrow (1 - r)\bar{\boldsymbol{\Lambda}}_2^{(k)} + r \sum_{i \in \mathcal{M}} \mathbf{k}_{\mathbf{u}i} \mathbf{k}_{\mathbf{u}i}^\top \mathbf{g}_{2,i}^{(k)}$                          *Natural gradient step*
12: **end for**
13: $\boldsymbol{\lambda}_{t+1} \leftarrow \boldsymbol{\lambda}^{(K)}$                                   *End of E-step*
14: $\boldsymbol{\theta}^{(0)} \leftarrow \boldsymbol{\theta}_t$                                   *Initialization of the gradient descent iterations*
15: **for** $s = 0 \dots S - 1$ **do**
16:     $\tilde{\mathcal{L}}^{(s)}(\boldsymbol{\theta}) = \log \mathcal{Z}^{(s)}(\boldsymbol{\theta}) + c^{(s)}(\boldsymbol{\theta})$                                   *ELBO*
17:     $\boldsymbol{\theta}^{(s+1)} \leftarrow \boldsymbol{\theta}^{(s)} + \gamma \nabla_{\boldsymbol{\theta}} \tilde{\mathcal{L}}^{(s)}(\boldsymbol{\theta})$                                   *Gradient step for $\boldsymbol{\theta}$*
18: **end for**
19: $\boldsymbol{\theta}_{t+1} \leftarrow \boldsymbol{\theta}^{(S)}$                                   *End of M-step*

# F    Data Sets and Experimental Details

## F.1    UCI Data Sets

For the regression experiments, we ran the E-step with a learning rate of 1. The update amounts to a closed form GP regression step given we have a conjugate model. We then ran the M-step 15 iterations with a learning rate of 0.2. In the classification examples we do not have closed form updates and so ran the E-step 8 times with a learning rate of 0.7. The M-step was ran the same way as in regression experiments. All other specifications where the same in all experiments. We choose $m = 50$ and given the data sizes were small, we set the mini batch to equal the data size $m_b = n$, so non stochastic gradients. The inducing points were initialized by K-means and optimized in the M-step along with hyper parameters. We ran all experiments a total of 20 full EM iterations. We ran 5-fold cross validation and in Fig. 3 plotted the mean result of the the folds. The kernel used was a Matérn-$5/2$ with lengthscale and amplitude both initialised at 1 similarly if a Gaussian likelihood was used it was likewise initialised to 1. We now detail each data set: **Airfoil**: The airfoil self-noise data set is regression task to predict scaled sound pressure. The data set has $d = 5$ and $n = 1503$ entries. **Boston housing**: The task is to predict the median value of owner-occupied homes. The data set has $d = 12$ and $n = 506$ entries. **Concrete**: The concrete compression data set is another regression experiment, where the goal is predict concrete compressive strength with $d = 5$ and $n = 1030$. **Sonar**: The data set is a classification example so we use a binomial likelihood. The goal is to predict from some sonar information if an object is a rock or a mine, the number of features is $d = 60$ and number of data points $n = 208$. **Ionosphere**: Another classification example where, 'Good' radar shows evidence of some type of structure in the ionosphere and "Bad" no evidence. The ionosphere data set has $n = 351$ and $d = 34$. **Diabetes**: The goal of the diabetes experiment is based on patient medical information can we predict the diabetic outcome. The data consists of $d = 8$ and $n = 768$ entries.

## F.2    MNIST Experiments

MNIST [1], available under CC BY-SA 3.0, is a handwritten digit classification task for digits 0–9. We used a softmax likelihood with 10 latent GPs, one for each digit. The data set is $n = 70,000$ and $d = 256$. We again used a Matérn-$5/2$ covariance function and set the number of inducing points $m = 100$ and used a minibatch size of $n_b = 200$. The kernel lengthscale $\ell$ and amplitude $\sigma^2$

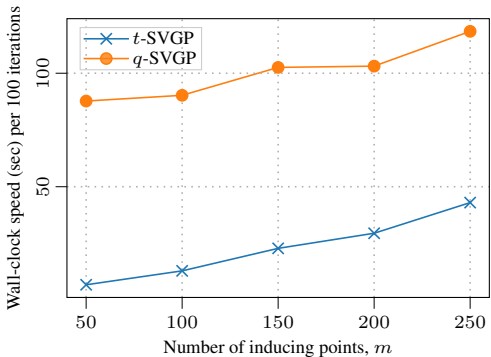

Figure 5: Wall-clock speed for $q$-SVGP and $t$-SVGP as a function of the number of inducing points $m$ on the MNIST experiment.

were both initialised to 1 and the inducing points were randomly initialised. We alternated between different learning rates and number of E and M-steps as detailed in Table 1.

### F.3 Illustrative Examples

For Fig. 1 (right) and Fig. 2 the experimental set up was similar. We considered a simplified one-dimensional GP classification task simulated by thresholding a noisy sinc function and simulating $n = 100$ observations. We considered $m = 10$ equally spaced inducing points for this task and fixed the lengthscale hyperparameter to $\ell = 1/2$.

### F.4 Additional Experiments

We include Fig. 5 to show the effect of changing the number of inducing points on the wall-clock speed. The experiment is the same as in App. F.2 but we now run only for 100 iterations of a single E and M step. The chart shows that there is a constant factor caused by our computationally cheaper E-step, the effect is substantial in most practical settings where $m$ is set below 250.

## G   Author Contributions

The idea of dual parameterization presented in the first part of Sec. 3 and the new lower bound discussed in Sec. 3.1 is due to MEK. The idea of using the dual parameterization to speed up SVGP was conceived by PEC and VA, who derived the bound, with inspiration from separate prior work by PEC, VA, and AS. PEC had the main responsibility of implementing the methods and conducting the experiments, and VA of formalizing the methods. All authors contributed to finalizing the manuscript.