# OpenReview forum: "Dual Parameterization of Sparse Variational Gaussian Processes"
_NeurIPS.cc/2021/Conference — NeurIPS 2021 Poster_

### Official Review · Reviewer_SLDR · 2021-07-14

**Rating:** 6
**Confidence:** 3

**Summary:**

This paper proposed a new dual parametrization method to improve the optimization efficiency for variational sparse Gaussian processes. The method speeds up inference than natural gradient descent and achieves a tighter lower bound for hyperparameter learning.

**Ethical Concerns:**

N.A.

**Limitations And Societal Impact:**

Focus more on the writing

**Main Review:**

Originality: This paper introduces a new method for optimizing the variational sparse Gaussian processes.

Quality: The authors should focus more on the technical writing to make them error-free:
Line 68: makes it a drawback .....
Line 130: can be approximated....
Line 519 (appendix)
......
Clarity: As mentioned above, the technical writing is not very sound and fluent which is misleading sometimes.

Significance: The experimental results generally demonstrated the technical part in the paper.

**Time Spent Reviewing:**

6

---

> ### Author Response · Authors · 2021-08-09
> **Reply to reviewer SLDR**
>
> We thank the reviewer for pointing out these unclear points. We will carefully go through the manuscript and improve these points for better readability and making it easier to understand the paper.

---

### Official Review · Reviewer_r5PB · 2021-07-16

**Rating:** 6
**Confidence:** 4

**Summary:**

The authors argue for a different (existing) parametrization for training (non-conjugate) Gaussian Processes.
They show that this parametrization can be interpreted as the optimal result of a dual problem, and that one can compute easily natural gradient updates.
Additionally this parametrization "may" lead to a tighter ELBO because of a stronger dependency on the hyperparameters (kernel parameters).
Experiments are run to show the improvement over the standard "q-SVGP".

**Limitations And Societal Impact:**

Authors mention limitations in Section 6, but there is no mention on how the method could fail.

**Main Review:**

The paper is well written and exposes clearly all the different elements to understand the theory and method introduced.
The main contributions are to bring a different view to the parametrization introduced in Csato 02' as an optimization problem that is mathematically interesting and derive natural gradient updates for this new parametrization.
Since the parametrization is already well known, the contribution is only limited and some of the mentioned advantages have flaws.

- There is a large flaw in the reasoning and the comparison with Salimbeni 18'. It is argued that  "[t-SVGP] is much cheaper than previous calculations of the natural gradient", because one supposedly does not have so many reparametrizations to work with. However $\nabla_\mu E_{q(f)}[\log p(y|f)]$ requires the same parametrization as Salimbeni 18' since gradients given the second expected parameter $(mm^\top + S)$ cannot be found directly (and therefore require a reparametrization as well)
-  The mention "It is a first order update" (L175)  sounds wrong since gradient derivatives given $S$ imply second order derivatives.
- Using natural gradients in Salimbeni 18' requires extra-care with the learning rate, especially in the first iterations. Some additional comments in this regard for t-SVGP would be good.
- It is not enough in the experiments to compare only to the standard q-SVGP, there should be at least a direct comparison with Opper 09', so one can see what the natural gradient approach actually brings.
- There are missing references for relevant work on this area, notably via the use of latent variable augmentation, see e.g. Wenzel 19' or more generally Galy-Fajou 20'
- Minor comments:
	- L100: "leads to a convex objective" Since the objective is defined to be the ELBO, it should be concave.
	- L192: "after one E-step" Repetition with the beginning of the sentence
- References
	- Csato 02' : Sparse on-line Gaussian processes, 2002, Csato and Opper
	- Opper 09' : The Variational Gaussian Approximation Revisited, 2009, Opper and Archambeau
	- Salimbeni 18' : Natural Gradients in Practice: Non-Conjugate Variational Inference in Gaussian Process Models, 2018, 		Salimbeni, Eleftheriom and Hensman
	- Wenzel 19' : Efficient Gaussian Process Classification Using Polya-Gamma Data Augmentation, 2019, Florian Wenzel, Theo Galy-Fajou, Christan Donner, Marius Kloft and Manfred Opper
	- Galy-Fajou 20' : Automated Augmented Conjugate Inference for Non-conjugate Gaussian Process Models, 2020, Théo Galy-Fajou, Florian Wenzel, Manfred Opper

**Time Spent Reviewing:**

5

---

> ### Author Response · Authors · 2021-08-09
> **Reply to reviewer r5PB**
>
> We thank the reviewer for their thoughtful review and detailed comments. We will respond to the comments in the order received.
>
> > Q1: There is a large flaw in the reasoning and the comparison with Salimbeni 18'
>
> Please see the response to reviewer XG5T for where the speed-up compared to Salimbeni 18’ comes from. The gradient of the variatiational expectations $VE = E_{q(f_n)} \[\log \,p(y_n|f_n)\]$ with respect to the expectation parameter of the variational distribution $\mu_u$ can be calculated by applying the chain rule. Indeed, the statistics of $q(f_n)$ depends directly on $[m_u, S_u]$ which itself depends on $\mu_u =  [m_u, S_u+m_u m_u^T]$. This is detailed in the supplementary material in Eq. (61).
>
> Our parameterization is not the same as Salimbeni’s, because we parameterize the posterior precision via the deviations to the prior precision $K^{-1}$, via sites. This parameterization ‘via deviation’ leads to simpler updates by bypassing the need to differentiate through the ELBO and the transformation between this parameterization and natural parameters.
>
> As noted, we need to transform Gaussian distributions with $m^3$ cost, to build the predictions $q(f_n)$ and during the chain rule from mean/chol_cov to expectation parameters. This is less than the number needed in the approach by Salimbeni (see response to Q1 of reviewer XG5T).
>
> > Q2: The mention "It is a first order update" (L175) sounds wrong since gradient derivatives given imply second order derivatives.
>
> We agree that the statement "It is a first order update" (L175) is slightly confusing. We meant it in the optimization sense, that is we do not compute derivatives beyond order 1 or a product thereof. An optimization using the Hessian to precondition the gradient would be a second order method in that sense. Our work exploits the known fact that natural gradient descent in exponential family distributions can be computed without the need to compute the Fisher information matrix, leading to a first order method. We amended the manuscript to reflect this.
>
> > Q3: Using natural gradients in Salimbeni 18' requires extra-care with the learning rate, especially in the first iterations. Some additional comments in this regard for t-SVGP would be good.
>
> We agree with the reviewer that the q-SVGP natural gradient is sensitive to the learning rate, especially in the stochastic gradient setting. Our method is more robust because our variational parameters constitute deviations to the prior, which for second order are added to the inverse of the well behaved prior covariance $K_{uu}$, which acts as a regularizer. In our experiments, we found we could set a much higher learning rate for our method t-SVGP than for the q-SVGP implementation, which is not visible in our experiment because we needed to tone down the learning rate for q-SVGP not to fail when comparing the methods.
>
> > Q4: It is not enough in the experiments to compare only to the standard q-SVGP, there should be at least a direct comparison with Opper 09', so one can see what the natural gradient approach actually brings.
>
> Our sparse parameterization is a direct extension of the seminal Opper 09’ paper. When $Z=X$, the optimal sites are univariate and we recover the Opper 09’ parameterization. The parameterization used in Opper 09’, however, does not lend itself easily to optimization. Khan et al. 2012 (Sec. 5) indeed showed that the resulting objective is non-convex and thus cannot be efficiently optimized via classic gradient descent—which the authors tried to do. In GPflow, the VGP implementation using the more costly mean/cholesky of covariance parameterization for the variational distribution is almost always preferred because the objective is then convex. However, both the GPflow parameterization and Opper 09’s VGP algorithm scale cubically with the number of data points.
>
> Our algorithm for $X=Z$ corresponds to Khan et al. 2017’s CVI method (who also provide a detailed discussion and comparison of Opper 09’). Most large scale ($n>1000$) applications of Gaussian processes models now resort to some kind of sparse approximation to speed up computations. Opper 09’ is today a theoretical inspiration more than a practical one.
>
>
>
> > Q5: There are missing references for relevant work on this area, notably via the use of latent variable augmentation, see e.g. Wenzel 19' or more generally Galy-Fajou 20'
>
> We are aware of the work by Galy-Fajou 20' and find it an original approach to approximate inference in GP models with a large class of non-conjugate likelihoods. Galy-Fajou 20'  introduces a new generative model (with additional variables, yet equivalent to the classic model once these are marginalized out) and techniques to perform approximate inference therein (mean field variational inference and Gibbs sampling).
>
> Our work is focused more on optimization with the dual parameterization and via mirror descent.
> As such, we have focused our review of past work more closely tackling these problems and our review of approaches to inference in GP models is not exhaustive. For example we omitted the large body of work on different inducing features, or alternative generative models.
>
> For completeness, we have now introduced Galy-Fajou 20' into our review of past work.

---

### Official Review · Reviewer_XG5T · 2021-07-16

**Rating:** 6
**Confidence:** 4

**Summary:**

The authors introduce an efficient natural gradient based approximate inference scheme for sparse variational Gaussian Process models. The approach leverages a "dual parameterization" that exploits Lagrange multipliers along the lines of "Decoupled variational Gaussian inference" (Khan 2014). These dual parameters are conceptually similar to site parameters in expectation propagation. In experiments it is shown that this dual parameterization scheme results in faster optimization and tighter bounds (although solid theory is lacking for the latter claim, at least for the non-conjugate case).

**Limitations And Societal Impact:**

The authors do not explicitly discuss potential negative societal impact of their work.

As suggested above, I think it would be valuable if the authors further delineated regimes (and verified empirically) in which they do and do not expect substantial speed-ups. In particular CPU versus GPU and few inducing points versus many inducing points.


**Main Review:**

Gaussian process (GP) models are popular due to their good uncertainty quantification, interpretability, and ability to include prior information through kernel design. Thus improvements to GP inference are likely to be of interest to an active if somewhat moderately sized NeurIPS subcommunity. This submission has pretty clear exposition and does a reasonably good job of positioning itself w.r.t. prior work. The experiments are moderately comprehensive and give some amount of confidence in the general claim that this method offers non-negligible inference speed-ups. The work is somewhat incremental, as it directly builds on other references, including e.g. various Khan et. al. and Hensman et. al. papers cited in the submission. Nevertheless, I think this work could be of interest to the NeurIPS community, since it offers concrete speed-ups over natural gradient approaches to SVGP as well as an overview/application of approaches like "Decoupled variational Gaussian inference" that, as the authors say, can be described as a "known, yet rarely used parameterization for Gaussian distributions."

Given the somewhat incremental nature of this work, I think the work could be considerably improved if the empirical evidence for inference speed-ups could be expanded. In particular as far as I can tell all the experiments use a pretty modest number of inducing points (100). Does the quoted 5x speed-up (line 322) persist for 500 or 1000 inducing points? Does the performance improvement disappear or largely disappear when implemented on GPU? As far as I can tell the experiments are all performed on CPU.

Additional comments/questions:
- “The formulation is fast since it avoids the use of sluggish automatic differentiation to compute the
natural gradients. “ (line 42) Can you clarify why this is exactly? As I understand it we are talking about a multiplicative constant and not an asymptotic speed-up. Has this more to do with the limitations/flexibility of current AD systems or is this something more fundamental?
- Typos: mirrordescent (x2), optimial

## After author response

I thank the authors for their response. Given the (somewhat) incremental nature of the work I maintain my original score. Nevertheless I would be in favor of seeing this submission accepted for publication, as I believe it could be of interest to the community. I suggest the authors include experiments with a larger number of inducing points. In most practical applications (unless one is particularly compute constrained) M ~250-500+ inducing points often deliver tangible performance improvements over much smaller numbers of inducing points. As such this represents a common and important regime.

**Time Spent Reviewing:**

1.2

---

> ### Author Response · Authors · 2021-08-09
> **Reply to reviewer XG5T**
>
> We thank the reviewer for their comments and suggestions.
>
> We claim two advantages for our method. The first relates to our choice of parameterization and states that it helps the optimization of hyperparameters. The second is that the natural gradient parameter updates have a cheaper computational complexity than the pre-existing method of Salimbeni et al.
>
> The reviewer’s comments are on this second point. The speed-up advantage between our implementation of natural gradient updates in Eq. (14) and that of Salimbeni 18’ Eq. (10), is not obvious and we here explain it further.
>
> > Q1: Does the quoted 5x speed-up (line 322) persist for 500 or 1000 inducing points?
>
> There are shared components in both our and Salimbeni’s method to perform an optimization step. Both require computing posterior predictions $q(f_n)$, variational expectations $VE = \sum_n E_{q(f_n)} \log\,p(y_n|f_n)$ and a $O(m^3)$ transformation of gradients with respect to some Gaussian parameters.
>
> The relative importance of these shared components, versus the ones individual to each method, in an optimization run depend on the number of inducing points $m$ and the number of data points in the training batch $n$. Typically, the bigger the $m$, the bigger the shared cost, thus the smaller the marginal benefit of our method. But as shown in our experiments the gains are substantial for settings of practical interest.
> > Q2: “The formulation is fast since it avoids the use of sluggish automatic differentiation to compute the natural gradients. “ (line 42) Can you clarify why this is exactly?
>
> Salimbeni’s method relies on reverse-mode automatic differentiation to back-propagate through the ELBO and through a transformation of the Gaussian parameterization Eq. (10) in Salimbeni 18’. Instead, our method only computes a forward mode derivative of each term of the VE.
>
> Our derivation shows that these extra steps in Salimbeni 18’ are unnecessary. Indeed by expressing mirror descent in the expectation parameters, the gradients of the KL divergence in the ELBO are available in closed form and directly lead to natural gradient updates in the natural parametrization which is the one we use. We thus skip these unnecessary gradient computations and change of parameterization which account for a large fraction of the total computational cost.
>
> In the code submitted with the manuscript, we reuse gpflow’s methods as much as we can, which leads at places to extra computations (for example, predictions could be made faster with custom code). Yet, our timing experiments still showcases speed gains.
>
> > Q3: As I understand it we are talking about a multiplicative constant and not an asymptotic speed-up.
>
> Yes, the speed up is in actual multiplicative constants in front of polynomials of $m$ and $n$, the most costly of which being $nm^2$ and $m^3$. Due the unnecessary computations highlighted in Q2, adding up all the operations, we end up with cheaper updates
>
> > Q4: Has this more to do with the limitations/flexibility of current AD systems or is this something more fundamental
>
> Reverse mode AD is an efficient formulation of the chain rule that bypasses the need to explicitly compute and store large Jacobians. When assessing the cost of Salimbeni’s update, we do take into account this efficient formulation. For instance, when comparing to Salimbeni 18’, we use the authors’ efficient jacobian vector product implementation turning a naive $O((m+m^2)^3)$ cost into a $O(m^3)$ cost. Nonetheless, by exploiting an alternative parameterisation of the variational problem, we avoid to compute gradients altogether, which mostly explains our computational savings.
>
> > Q5: Does the performance improvement disappear or largely disappear when implemented on GPU?
>
> We are currently exploring implementation on GPUs. Details are important when it comes to implementation and particular platforms might induce more overhead for one method than the other. But all things equal, GPU should a priori equally speed up both methods, which use the same linear algebra primitives, and leave the relative gains unchanged.

---

### Official Review · Reviewer_zj8L · 2021-08-01

**Rating:** 8
**Confidence:** 4

**Summary:**

This paper proposes to speed-up approximate inference in the sparse variational GP model using a dual parameterisation (similar to an EP approach) where each data point is assigned an additional dual variable that may be updated efficiently using natural gradient descent with reduced complexity on the natural gradient updates that also lead to a tighter bound on the evidence. The paper illustrates the efficacy of the approach over standard SVGP variational inference on a number of standard benchmark datasets and demonstrates good empirical performance whilst reducing training time.

**Ethical Concerns:**

No ethical concerns.

**Limitations And Societal Impact:**

Yes

**Main Review:**

I found the paper to be clear to read with a good coverage of the background material and justification for the proposed methodology. The authors highlight that the proposal brings together a number of approaches that are already established (e.g. SVGP, natural gradient descent and EP) but, in my opinion, this is not a limitation as the paper treats the material well, does not overclaim the novelty but instead makes the point that bringing these things together in this way yields a very satisfactory approach that is neither well-known nor well used (to the best of my knowledge). That it performs competitively in established code-bases (such as GPflow) is a good indication of importance of the result and its general applicability. In particular, the recognition that improving the calculations of the natural gradient updates is probably the most satisfying part of the paper and the case for this is well made.

Overall, I believe that the paper provides sufficient empirical evidence to support the theoretical ideas behind the proposed method. I can see that there might be objections to datasets such as MNIST but I agree that it is the scaling we are interested in here and the separation of the objective wrt hyperparameters and variational params provides a strong motivation and good result for the paper.

To the best of my knowledge the approach described has not been explicitly presented before and I am in favour of accepting the paper.

Other comments:

- Thinking about fair comparison for the best q-SVGP - a number of papers have questioned whether the alternation approach of EM does always lead to improved convergence (i.e. is coordinate ascent always better than something like a Gauss-Newton approach that approximates the off-diagonal blocks of the Hessian somehow). I wonder if the authors could comment upon whether or not they have tried these sorts of approaches? I can well believe that the natural gradients lead to better conditioning (effectively the appropriate blocks of the Hessian for approx second-order optimisation) but, if I understand correctly, they will assume “zero blocks” in the Hessian in the cross terms between, say, the natural gradient parameters and the hyper parameters. My understanding is that these are usually dealt with first order approaches with momentum, etc.. but I don’t know if there has been any more in-depth investigation for this approach? E.g. something like L-BFGS or similar?

- Could the authors avoid using “e” as a parameter (e.g. (11)) since its also used as the exponential (e.g. (12))?

-  Typos: lines 81 (-them by-), 91 (_integrals_), 110 (Gradient Descent?)

**Time Spent Reviewing:**

4

---

> ### Author Response · Authors · 2021-08-09
> **Reply to reviewer zj8L**
>
> We thank the reviewer for their considerate review and appreciate the thoughtful comments. We have fixed the typos that were pointed out, and will address the remaining comments below..
>
> > On whether coordinate ascent always better than something like a Gauss-Newton approach that approximates the off-diagonal blocks of the Hessian somehow
>
> Whether a coordinate ascent method, that separates the parameters of the variational distribution $q$ from the hyperparameters $\theta$, is always better than second order methods applied to both parameters $\(q, \mathbf{\theta}\)$ is an important question. By using a coordinate ascent scheme, we align with the original EM algorithm. This scheme forces the $q$ distribution to stay close to the optimal variational distribution $q^*$ when the hyperparameters are optimized. When jointly optimizing, this property is lost. Whether this matters in practice, is an empirical question.
>
> Salimbeni 18’ (Sec. 5.3) partially answered this question by noting that coordinate ascent always outperforms a momentum based Adam optimizer used to jointly optimize $q$ and $\theta$. The coordinate ascent scheme is what allows us to exploit the geometry of $q$ to derive efficient natural gradient updates. This computational advantage would be lost if directly using alternative 2nd order optimization on both $q$ and $\theta$ ignoring this geometry. Additionally, Salimbeni 18’ (Sec. 5.1) showed that natural gradient updates outperform L-BFGS (Hessian based) when only optimizing $q$ (fixed theta). Usually the number of hyperparameters $\theta$ is much smaller than the number of variational parameters, so exploiting the structure of $q$ is essential to obtain performance gains.
>
> In summary, previous evidence suggests that coordinate ascent has an advantage over joint optimization. By improving over the pre-existing natural gradient algorithms, we widen this performance gap in favor of coordinate ascent. We will attempt to highlight these aspects and also point the interested reader to the experiments in Salimbeni 18’.
>
> References:
>
> [Salimbeni 18’]: Salimbeni, H., Eleftheriadis, S., & Hensman, J. Natural gradients in practice: Non-conjugate variational inference in Gaussian process models. In AISTATS 2018.

---

### Official Review · Reviewer_aCDy · 2021-08-08

**Rating:** 6
**Confidence:** 3

**Summary:**

The paper revisits a known, but seemingly forgotten dual parametrization for multivariate Gaussian distributions that naturally lends itself to variational inference in Gaussian process models. This dual parametrization was introduced in previous works for variational Gaussian inference using a Lagrangian relaxation of the ELBO loss. Similarly to commonly used parametrizations, the dual parametrization can be combined with natural gradients based optimization that allows for faster training of the variational distribution by exploiting the information geometry of distribution space. However, the required natural gradients are considerably cheaper to compute in this setting due to only requiring first-order gradients. This parametrization is then incorporated into a sparse variational inference scheme for GPs. One of the computational difficulties when training the the SVGP lies with the coupled optimization of the variational distribution and the prior hyperparameters, for which several heuristics have been introduced in the community. The experiments compare the proposed parametrization for SVGP with the commonly used whitened one. They demonstrate that for small and mid-sized datasets, gradient descent along the trajectory of optimal variational distributions results in more stable learning of the hyperparameters. For larger datasets, the coupled stochastic optimization of the two (with only partial optimization of the variational distribution) leads the dual formulation to equivalent or slightly better optima measured in terms of predictive density, but it is considerably faster than the baseline due to cheaper natural gradients.

**Ethical Concerns:**

No concerns.

**Limitations And Societal Impact:**

At the moment there is no discussion of limitations, which may be useful to include with the experimental evaluation. In particular, if there are any drawbacks of the proposed parametrization compared to the standard one would be useful to know. It is also not clear to me if the discussed dual parametrization limits the space of approximating distributions, or if any multivariate Gaussian can be represented in this form as well. Also, did the authors face any challenges during the implementation of the proposed model, or during its optimization?

**Main Review:**

This is a well-written paper that recycles a previously known parametrization for variational Gaussian inference that has not been employed in the SVGP context before. The main idea is therefore relatively simple, which in concept follows from simply putting these together. However, in my view, there is considerable value in this due to the nontrivial insight that is required to carry out the required calculations, put them into an implementation and evaluate them experimentally. Especially so, because the main problem that the paper is trying to address (optimization in SVGP) is of considerable significance, and the results are of interest. Judging purely by the results, the approach does seem promising and I think this paper can be a valuable contribution to the GP community.

One obstacle I see to these ideas becoming adopted and their use more wide-spread is that the description of the proposed approach is not very clear at the moment. In particular, Section 3 is hard to follow because no intuition is provided about the dual parametrization, and only a brief survey is given (although with several references), which is not an adequate initiation to the approach for those not already familiar with it. Given that the main tenet of the paper is that this parametrization seems to be unexplored in the SVGP context, I would have appreciated if a more detailed description was given. In contrast, Section 2 spends about two pages on recapping well-known ideas. Some of this space could probably be reallocated to the description in Section 3. This also makes Section 4 somewhat hard to follow, where the idea is combined with the SVGP. Section 5 on the other hand is very clear, the experimental evaluation and the results are interesting.

Overall, although neither of the core ideas in the paper are novel, I believe that a novel combination of known ideas can also be a valuable contribution. However, my main criticism of the paper is regarding the presentation of the idea that they are trying to introduce, which in its current form might limit their use due to lack of clarity. If the authors can improve on the presentation in these sections, I would be inclined to slightly raise my rating.

**Time Spent Reviewing:**

6

---

> ### Author Response · Authors · 2021-08-09
> **Reply to reviewer aCDy**
>
> We thank the reviewer for their comments and overall appreciation of the work regarding 1) the *non-trivial* nature of the insight to put together known concepts, 2) the *considerable* significance of the problem to improve inference and learning in SVGP models, and 3) the quality of the manuscript deemed *well-written*. We feel the same way that our work brings together known concepts to provide a better parameterization for SVGP along with a faster and better behaved optimizer.
>
> > Q1: On the clarity of the description of the dual parameterization
>
> Our contribution could be summarized as performing a particular instance of mirror ascent on the ELBO: with a KL regularizer and using the expectation parameterization for $q(u)$. This choice of optimization leads to natural gradient updates with the structure of dual parameterization.
>
> It took more than a decade of work to reach this simple formulation and in this article we attempt to meticulously review and credit the various steps taken along the way. We explain how the parameterization has arisen via the tools of Lagrangian duality (though without leading to the simple optimization introduced here) and how it corresponds to a parameterization used in Expectation Propagation, where it lead to efficient algorithms albeit arguably not as versatile as those developed within the framework of variational inference.
> This plurality of origins is reflected in how we interchangeably refer to the dual parameterization or to sites.
>
>
> > Q2: On structure of the paper shortening section 2, extending section 3 and making section 4 easier to follow.
>
> We agree with the reviewer on this point. Given the topic touches on many areas we wanted to ensure we gave the reader a brief tour. However, as you point out a large fraction of Section 2 is background and can be moved to the appendix. This space together with the additional space in the camera-ready submission can be used in Section 3 to beef up the intuition of the dual parameterization.
>
> Our plan is to amend the manuscript to clearly state that the purpose of Section 3 is to introduce the structure of the parameterization and to review the past work that motivated its introduction. We will also articulate better the key point of our manuscript which is that we obtain simple and efficient updates for this parameterization which were lacking up to now and that we obtain a better bound for hyperparameter optimization.
>
> We also then believe there will be more space to elaborate on our Section 4 of the particular nuances around adapting the dual parameterization to the sparse and stochastic GP setting.
>
> > Q3: On discussing the limits of the dual parameterization
>
> The dual parameterization for $q(u)$ is a subset of the set of multivariate normal distributions. However, in a similar vein as in Opper 09’ we show that this parameterization contains the optimal variational distribution. It is thus not restrictive.
>
> By parameterizing deviations from the prior precision, this parameterization is more stable numerically than the direct parameterization of the moments of $q(u)$ as is done in Salimbeni 18’. The learning rate can be set to higher values without leading to the ‘Cholesky errors’ that usually happen when the optimization steps lead to badly conditioned second order statistics.
>
> Finally, at the core, the statistics of the ‘sites’ of the dual parameterization can be interpreted as local Gaussian likelihood approximations, and can easily be constrained to be valid in settings where the natural gradient updates may set them otherwise (for example when the likelihoods are not log-concave which would cause invalid updates).

---

> > ### Comment · Reviewer_aCDy · 2021-08-31
> > **Questions cleared**
> >
> > I thank the authors for their detailed answers to my points. The authors have adequately answered my questions, and my main criticism was regarding the clarity of the presentation in Sections 3-4. The authors said they would expand on these sections. As this would probably not be a major revision, I do not think it should be a cause for rejection, especially because the other reviewers found the presentation clear. However, I would like to emphasize I still think that it would be beneficial to add more details to help navigate through these sections those that are not that familiar with this branch of the literature.
> >
> > Other than that, I am mainly on the same viewpoint as Reviewer XG5T, namely that although the work is incremental in nature, it should definitely be of interest to the GP community, and hence I have updated my rating to 6 to support acceptance.

---

### Decision · Program_Chairs · 2021-09-27

**Decision:**

Accept (Poster)

**Comment:**

This paper revisits a known alternative, so-called dual, parameterization for models with Gaussian priors and iid likelihoods and applies it to stochastic variational inference in Gaussian process models. It shows _empirically_ that this parameterization results in faster optimization and tighter bounds that improve hyper-parameter learning. Results are shown on small UCI datasets and on MINST (n=70,000 datapoints).

Overall, the reviewers believe that the proposed approach is incremental as it takes a known idea (that of the dual parameterization) and applies to the SVGP context. However, all the reviewers agree that the paper provides a significant contribution to the NeurIPS community and that NeurIPS will benefit from knowing the details and findings in this paper.

Besides its incremental nature, one of the major criticisms of this paper was with respect to its clarity (Reviewer aCDy), which the authors seemed to have addressed satisfactorily. Another important point (raised by reviewer XG5T) concerns whether the benefits of the approach would extend to larger problems for which one would need a higher number of inducing variables ($m>100$). The authors have clarified that the bigger the $m$ the smaller the marginal gain obtained by the proposed method. However, the proposed approach still provides benefits for settings of practical interest.